# H3.3K27M mutant proteins reprogram epigenome by sequestering the PRC2 complex to poised enhancers

Dong Fang[1†‡], Haiyun Gan[1†], Liang Cheng[2], Jeong-Heon Lee[2], Hui Zhou[1], Jann N Sarkaria[3], David J Daniels[4], Zhiguo Zhang[1]*

[1]Department of Pediatrics and Department of Genetics and Development, Institute for Cancer Genetics, Irving Cancer Research Center, Columbia University, New York, United States; [2]Department of Biochemistry and Molecular Biology, Mayo Clinic, Rochester, United States; [3]Department of Radiation Oncology, Mayo Clinic, Rochester, United States; [4]Department of Neurosurgery, Mayo Clinic, Rochester, United States

**\*For correspondence:**
zz2401@cumc.columbia.edu

[†]These authors contributed equally to this work

**Present address:** [‡]Life Sciences Institute, Innovation Center for Cell Signaling Network, Zhejiang University, Hangzhou, China

**Competing interests:** The authors declare that no competing interests exist.

**Abstract** Expression of histone H3.3K27M mutant proteins in human diffuse intrinsic pontine glioma (DIPG) results in a global reduction of tri-methylation of H3K27 (H3K27me3), and paradoxically, H3K27me3 peaks remain at hundreds of genomic loci, a dichotomous change that lacks mechanistic insights. Here, we show that the PRC2 complex is sequestered at poised enhancers, but not at active promoters with high levels of H3.3K27M proteins, thereby contributing to the global reduction of H3K27me3. Moreover, the levels of H3.3K27M proteins are low at the retained H3K27me3 peaks and consequently having minimal effects on the PRC2 activity at these loci. H3K27me3-mediated silencing at specific tumor suppressor genes, including Wilms Tumor 1, promotes proliferation of DIPG cells. These results support a model in which the PRC2 complex is redistributed to poised enhancers in H3.3K27M mutant cells and contributes to tumorigenesis in part by locally enhancing H3K27me3, and hence silencing of tumor suppressor genes.
DOI: https://doi.org/10.7554/eLife.36696.001

## Introduction

Diffuse intrinsic pontine glioma (DIPG) is the most aggressive primary malignant brain tumor found in children (*Jones and Baker, 2014*). The median survival time after diagnosis is approximately one year with no cure in sight (*Dellaretti et al., 2012*). Recent studies have identified, in more than 75% of DIPG cases, a somatic mutation of the *H3F3A* gene, leading to a lysine 27 to methionine mutation at histone H3 variant H3.3 (H3.3K27M) (*Castel et al., 2015*; *Schwartzentruber et al., 2012*; *Sturm et al., 2012*; *Wu et al., 2012*, *2014*). In addition, *HIST1H3B* or *HIST1H3C*, one of the 13 genes encoding canonical histone H3.1/H3.2, is also mutated, resulting in the same K to M change in a small fraction of DIPG tumors (*Castel et al., 2015*; *Fontebasso et al., 2014*; *Solomon et al., 2016*). However, it remains largely unknown how the histone H3K27M mutation drives epigenetic changes and tumorigenesis.

H3.3 is a histone H3 variant that differs from H3.1/H3.2 by only four or five amino acid residues (*Campos and Reinberg, 2009*; *Maze et al., 2014*; *Szenker et al., 2011*). Compared to canonical histone H3.1/H3.2 that is assembled into nucleosomes in a replication-coupled process, histone variant H3.3 is assembled into nucleosomes in a replication-independent manner (*Ahmad and Henikoff, 2002a*; *Tagami et al., 2004*). H3.3 has diverse functions including epigenetic memory (*Ng and Gurdon, 2008*), heterochromatin formation (*Banaszynski et al., 2013*; *Rai et al., 2014*), silencing of endogenous retroviral elements (*Elsässer et al., 2015*), and gene transcription (*Ahmad and*

*Henikoff, 2002b*; *Chen et al., 2013*; *Deaton et al., 2016*). Genome-wide analysis of H3.3 distribution reveals that H3.3 is enriched at gene promoter and gene body of actively transcribed genes compared to lowly expressed genes. Moreover, H3.3 is also localized at gene regulatory elements including promoters and enhancers. The distinct distributions of H3.3 likely contribute to the diverse roles of H3.3 in cells.

H3K27 can be mono-, di- and tri-methylated (H3K27me1/me2/me3). H3K27me2/me3 is catalyzed by the Polycomb Repressive Complex 2 (PRC2) and plays an important role in the repression of developmentally regulated genes during development (*Banaszynski et al., 2013*; *Cao et al., 2002*; *Margueron and Reinberg, 2011*; *Simon and Kingston, 2013*). H3K27me3 is enriched at promoters of silenced genes. Moreover, low levels of H3K27me3 are also detected at poised enhancers along with H3K4me1 in mouse and human embryonic stem (ES) cells (*Creyghton et al., 2010*; *Rada-Iglesias et al., 2011*) as well as bivalent chromatin domains containing H3K4me3 (*Bernstein et al., 2006*). The PRC2 complex consists of four core subunits (Ezh2, Suz12, EED, and RbAP46/48), and methylates preferentially nucleosomal histone H3K27.

We and others have found that H3K27me3 levels are dramatically reduced in DIPG cells as well as other tested cell types or organisms expressing H3.3K27M mutant proteins, including mice and *Drosophila* (*Bender et al., 2013*; *Chan et al., 2013a*, *2013b*; *Funato et al., 2014*; *Herz et al., 2014*; *Lewis et al., 2013*). However, it remains in a debate on how H3.3K27M mutant proteins lead to a global reduction of H3K27me3 in cells. Several studies support a model that H3.3K27M mutant proteins may trap the PRC2 complex and lead to a global reduction of H3K27me3. For instance, it has been shown that, *in vitro*, PRC2 complex binds to H3K27M peptides with high affinity (*Jayaram et al., 2016*), and H3K27M peptides or mononucleosomes inhibit the enzymatic activity of PRC2 complex (*Jayaram et al., 2016*; *Jiao and Liu, 2015*; *Justin et al., 2016*; *Lewis et al., 2013*). Moreover, Ezh2 is enriched at H3.3K27M mononucleosomes compared to wild type H3.3 mononucleosomes isolated from cells (*Bender et al., 2013*; *Chan et al., 2013a*). However, this model has been challenged by several recent observations. For instance, using ChIP-seq, it has been shown that H3.3K27M mutant proteins co-localize with active histone mark H3K27ac where the PRC2 complex should be absent (*Piunti et al., 2017*). Moreover, the PRC2 complex binds H3K27M mononucleosomes with equal affinity as wild type nucleosomes (*Herz et al., 2014*). Therefore, additional studies are needed to help resolve the current debate.

In addition to the global loss of H3K27me3, we and others observed that H3K27me3 is retained or elevated at the promoters of a small coterie of genes in DIPG xenograft cell lines harboring H3.3K27M mutation (*Bender et al., 2013*; *Chan et al., 2013a*). Moreover, gene etiology analysis indicates that these genes are enriched in cancer pathways. We proposed that this locus-specific gain/retention of H3K27me3 acts together with the more general decline of H3K27me3 to promote tumorigenesis (*Chan et al., 2013a*). Supporting this model, it has been shown that Ezh2 is required for tumorigenesis of DIPG in a mouse model (*Mohammad et al., 2017*; *Piunti et al., 2017*). It has been shown that Ezh2 and H3K27me3 silence the tumor suppressor gene (TSG) p16 (*Cordero et al., 2017*; *Mohammad et al., 2017*; *Piunti et al., 2017*). However, it is unknown how H3K27me3 at these genes such as p16 is retained or gained in cells expressing H3.3K27M mutant proteins and whether other TSGs are also silenced via H3K27me3-mediated mechanism.

Here, we report a surprising observation that the PRC2 complex is sequestered at poised enhancers, but not at active promoters containing high levels of H3.3K27M mutant proteins. We propose that the poised enhancers but not active promoters possess a mechanism for the initial recruitment of the PRC2 complex. Once recruited, the H3.3K27M mutant proteins will help sequester the PRC2 complex to these sites, which reduces the availability of the PRC2 complex at other sites and thereby leads to a global reduction of H3K27me3. Moreover, the phenomenon of sustained or enhanced accumulation of H3K27me3 is detected in DIPG xenograft cell lines and one primary tumor sample, harboring the H3.3K27M mutation. Further, H3.3K27M mutant proteins associated with these H3K27me3 are low or absent, consequently having little effects on the activity of the PRC2 complex at these sites. Finally, we report that H3K27me3-mediated silencing of Wilms Tumor 1 (WT1) gene supports the proliferation of DIPG cells.

# Results

## H3.3K27M mutant proteins are enriched at highly expressed genes in both DIPG cells as well as mouse ES cells knocked-in with the H3.3K27M mutation

To understand how the reduction of H3K27me3 occurs in DIPG cells, SF7761 and SF8628, which contain a heterozygous mutation at *H3F3A* gene, replacing H3.3 lysine 27 with methionine (K27M), we analyzed the localization of H3.3K27M mutant proteins using H3K27M-specific antibody (*Figure 1—figure supplement 1A*). The H3.3K27M mutant proteins were enriched at actively transcribed genes compared to lowly expressed genes in both SF7761 and SF8628 cells (*Figure 1A and B*), a pattern that is similar to the localization pattern of wild type H3.3 proteins detected in other cell lines (*Banaszynski et al., 2013*). Under the same conditions, ChIP-seq signals in human neural stem cells (NSCs) with wild type H3 were not detected using the same H3K27M antibodies (*Figure 1C*), supporting the idea that H3.3K27M ChIP-seq signals detected in SF7761 and SF8628 are specific.

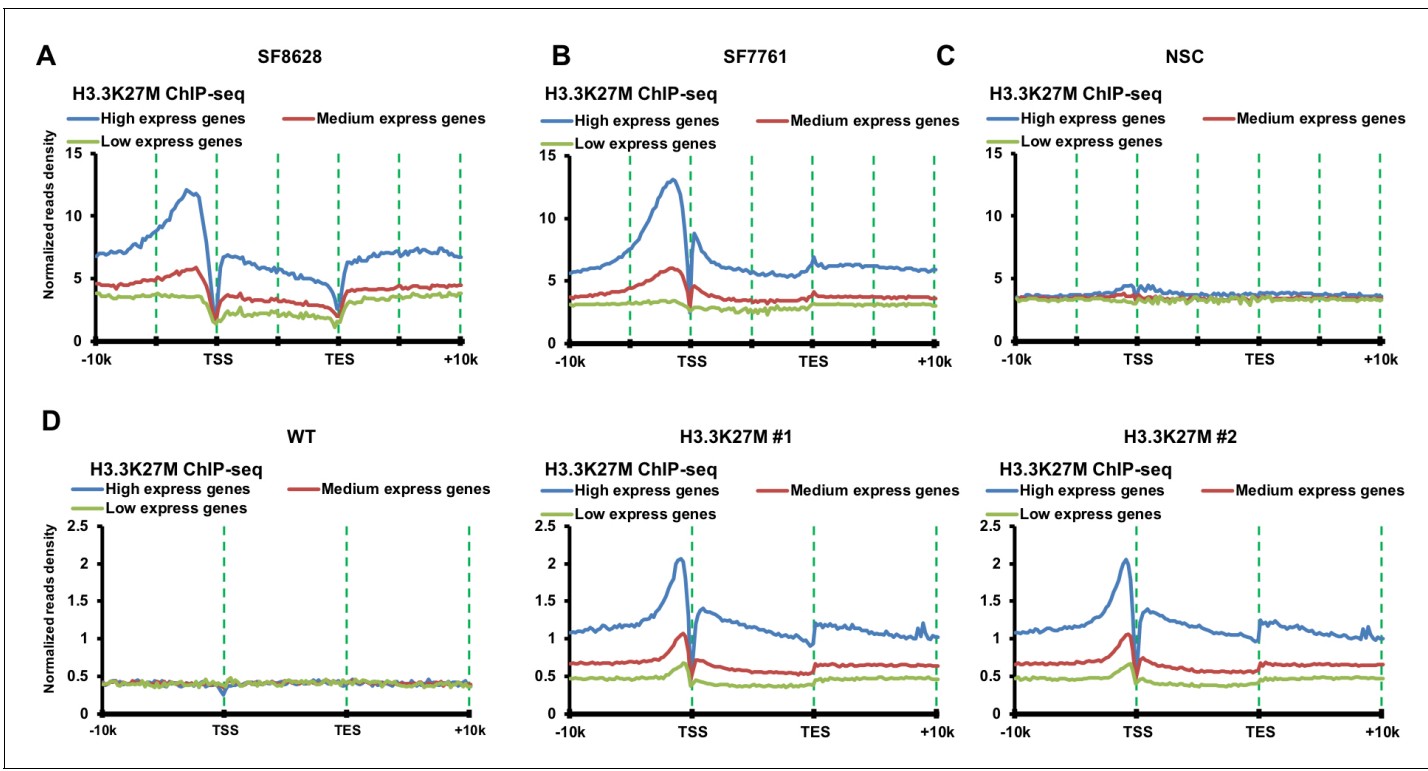

**Figure 1.** H3.3K27M mutant proteins are enriched at highly transcribed genes compared to lowly expressed genes in DIPG cells and mouse ES cells with H3.3K27M mutation. (A–C) H3.3K27M mutant proteins are enriched at highly transcribed genes compared to lowly expressed genes in DIPG cells. The average read density of H3.3K27M ChIP-seq in two H3.3K27M mutant lines SF8628 (A) and SF7761 (B), and reference human neuro stem cells (NSC, C) with wild type H3.3 from 10 Kb upstream of TSS to 10 Kb downstream of TES is calculated. The read density was normalized to Reads Per Kilo-base per 10 million mapped reads. The entire human genes were split into three groups according to their expression levels in the corresponding cell lines: highest expressed genes, medium expressed genes, and low expressed genes. (D) H3.3K27M mutant proteins are enriched at highly transcribed genes compared to lowly expressed genes in mouse ES cells. The experiments were performed as described in (A). The entire mouse genes were split into three groups according to their expression levels in wild type mouse ES cells using mouse ES cell gene expression dataset at GEO (GSE8024): high express genes, medium express genes, and low express genes.

DOI: https://doi.org/10.7554/eLife.36696.002

The following figure supplement is available for figure 1:

**Figure supplement 1.** Site-specific mutation at the *H3F3A* gene, resulting in expression of the H3.3K27M proteins in mouse ES cells does not change cell identity.

DOI: https://doi.org/10.7554/eLife.36696.003

To gain further insight into the reduction of H3K27me3 by H3.3K27M mutant proteins, we introduced the same heterozygous mutation at the *H3F3A* gene in mouse ES cells using the CRISPR/Cas9 genome editing technology and isolated two independent clones (*Figure 1—figure supplement 1B*). We chose mouse ES cells because it is relatively easier to introduce the site-specific mutation. Moreover, the localizations of H3.3 and H3K27me3 have been studied extensively before (*Banaszynski et al., 2013*; *Goldberg et al., 2010*). The two H3.3K27M mutant mouse ES cells grew like wild type mouse ES cells and had normal karyotype (*Figure 1—figure supplement 1C and D*). To analyze the impact of H3.3K27M mutation on H3K27 methylation, we compared the levels of H3K27me1, H3K27me2 and H3K27me3 in wild type and H3.3K27M mutant cells using different amount of extracts (*Figure 1—figure supplement 1E*). Compared to wild type cells, H3K27me1 was not affected to a detectable degree in H3.3K27M mutant cells under all these conditions, consistent with published study in DIPG cells (*Bender et al., 2013*). In contrast, H3K27me2 and H3K27me3 levels, while detectable in high amounts of cell extracts, were reduced dramatically in H3.3K27M mutant cells compared to wild type cells. Moreover, the reduction of H3K27me2/me3 in each of the H3.3K27M mutant ES cell clone is unlikely due to the reduced levels of the PRC2 complex (*Figure 1—figure supplement 1F*).

We next analyzed the chromatin distribution of H3.3K27M using ChIP-seq in wild type and these H3.3K27M mutant mouse ES clones. H3.3K27M ChIP-seq identified 29,424 and 27,048 peaks in H3.3K27M mutant clone #1 and #2, respectively, of which 22,959 peaks overlapped. About 4,555 H3.3K27M false-positive peaks were identified in wild type mouse ES cells and few of these peaks overlapped with H3.3K27M ChIP-seq peaks identified in H3.3K27M mutant cells (*Figure 1—figure supplement 1G*), further demonstrating the specificity of H3K27M antibodies. H3.3K27M mutant proteins in H3.3K27M mutant ES cells were also enriched at promoters and gene bodies of actively transcribed genes compared to lowly expressed genes (*Figure 1D*), a pattern that resembles wild type H3.3 in mouse ES cells (*Goldberg et al., 2010*). Together, these results indicate that H3.3K27M mutant proteins are high at the promoters of actively transcribed genes in both DIPG lines and mouse ES cells with the H3.3K27M mutation. Because the PRC2 complex in general is not localized at highly expressed genes, these results support the idea that the PRC2 complex is unlikely trapped at places with high levels of H3.3K27M mutant proteins and high gene activities (*Piunti et al., 2017*).

## Ezh2 is redistributed to poised enhancers without methylating H3K27 at these sites in mutant mouse ES cells

To understand how the expression of H3.3K27M mutant proteins results in reduced levels of H3K27me3, we first analyzed Ezh2 and H3K27me3 ChIP-seq peak numbers in wild type and H3.3K27M mutant mouse ES cells, an analysis not feasible for DIPG cells due to a lack of matched controls. We identified 30,295 H3K27me3 ChIP-seq peaks in wild type mouse ES cells, 17,158 peaks in H3.3K27M mutant clone #1 and 16,316 in H3.3K27M mutant clone #2 (*Figure 2—figure supplement 1A*). Almost all H3K27me3 ChIP-seq peaks in each H3.3K27M mutant lines overlapped with those identified in wild type mouse ES cells (*Figure 2A* and *Figure 2—figure supplement 1A*). Moreover, compared to wild type mouse ES cells, H3K27me3 levels at all of these peaks in H3.3K27M mutant lines were reduced (*Figure 2—figure supplement 1B*). These results indicate that almost all H3K27me3 peaks detected in H3.3K27M mutant mouse ES cells are PRC2 sites in wild type cells, but exhibit reduced levels of H3K27me3 compared to wild type mouse ES cells.

Surprisingly, we observed a dramatic increase of the Ezh2 ChIP-seq peak numbers in each of the H3.3K27M mutant clones. Specifically, 40% of the overlapped Ezh2 peaks between two H3.3K27M mutant clones were unique in H3.3K27M mutant mouse ES cells, and were not detected in wild type mouse ES cells using the same cut off. In contrast, the number of H3K27me3 peaks, majority of which overlapped with those in wild type ES cells, is far less in H3.3K27M mutant ES cells (*Figure 2A*). This result suggests that Ezh2 proteins, most likely the PRC2 complexes, are redistributed to new locations but do not generate new H3K27me3 peaks in H3.3K27M mutant mouse ES cells.

To gain insights of the properties of Ezh2 ChIP-seq peaks in H3.3K27M mutant mouse ES cells, we first separated 6,658 Ezh2 ChIP-seq peaks identified in H3.3K27M mutant cells into two groups, peaks commonly found in both wild type and H3.3K27M mutant ES cells and peaks unique to H3.3K27M mutant ES cells, based on their overlap with Ezh2 ChIP-seq peaks found in wild type ES cells (*Figure 2B*). First, we compared the levels of H3.3K27M and H3K27me3 at these two groups of

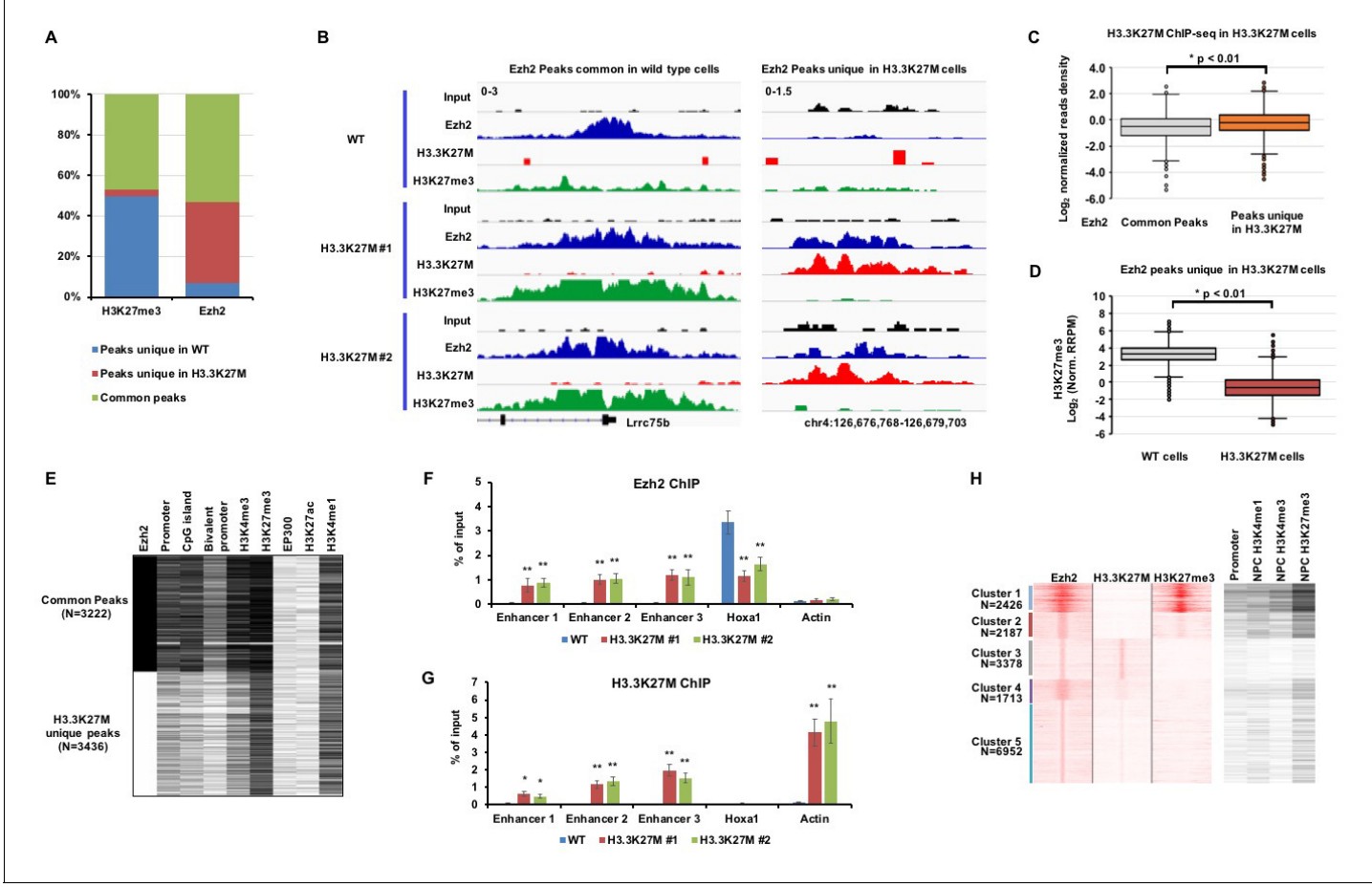

**Figure 2.** Ezh2 is sequestered at poised enhancers with H3.3K27M mutant proteins. (**A**) A large fraction of Ezh2 peaks unique in H3.3K27M mutant mouse ES cells was identified. The percentage of peaks (Y-axis) in each category (WT unique peaks, H3.3K27M unique peaks, and overlap peaks) of H3K27me3 and Ezh2 ChIP-seq (X-axis) is shown as indicated. The overlapped H3K27me3 and Ezh2 ChIP-seq peaks from two independent H3.3K27M mutant clones were used for the analysis. Peaks unique in WT: ChIP-seq peaks only identified in WT mouse ES cells but not in each of the two H3.3K27M clones. Peaks unique in H3.3K27M: the peaks are present in both H3.3K27M clones but not in WT mouse ES cells. Common peaks: the peaks present in both WT and two H3.3K27M clones. (**B**) The Integrative Genomics Viewer (IGV) tracks showing input, Ezh2, H3.3K27M, and H3K27me3 distributions of common Ezh2 peaks and Ezh2 peaks unique in H3.3K27M as in (**A**) in each of the two H3.3K27M mutant mouse ES clone. (**C**) Boxplot showing that H3.3K27M mutant proteins are low at common Ezh2 peaks compared to peaks unique in H3.3K27M. P value was calculated by two-tailed Student's t test. (**D**) At the Ezh2 peaks unique to H3.3K27M mutant cells, H3K27me3 levels in H3.3K27M mutant cells are low compared to wild type cells. P value was calculated by two-tailed Student's t test. (**E**) Heatmap showing genomic annotation of two groups of Ezh2 peaks. The common peaks: Ezh2 peaks present both in WT and H3.3K27M cells. H3.3K27M unique peaks: Ezh2 peaks only present in both two H3.3K27M clones. The peak number in each cluster (**N**) is indicated. Genomic annotation of each Ezh2 ChIP-seq peaks was performed by Perl (https://www.perl.org/). Black and white represents the presence and absence of the annotated element or histone marks, respectively. Promoter is defined as 2 Kb upstream and 1 Kb downstream of TSS. Bivalent promoters are downloaded from BGDB database (http://dailab.sysu.edu.cn/bgdb/). The histone mark and EP300 ChIP-seq datasets of mouse ES cells are from ENCODE (https://www.encodeproject.org/). (**F–G**) Ezh2 and H3.3K27M were enriched at the poised enhancers. Ezh2 (**F**) and H3.3K27M (**G**) ChIP-PCR results were shown. The enrichment of Ezh2 and H3.3K27M at three poised enhancers, labeled as enhancer 1 to 3, and one strong PRC2 site, Hoxa1, was analyzed by ChIP-PCR. Actin was used as the negative control for the Ezh2 ChIP and Hoxa1 was used as the negative control for the H3.3K27M ChIP. Data are mean ± SD (N = 3 independent replicates, *p<0.05, **p<0.01). (**H**) Heatmaps showing unsupervised clustering analysis of Ezh2 peaks for the overlap with H3.3K27M and H3K27me3 ChIP-seq in SF8628 as well as for their overlap with promoters, H3K4me1, H3K4me3 and H3K27me3 at NPCs. The histone ChIP-seq datasets for neuro precursor cells (NPC) are from ENCODE (https://www.encodeproject.org/).

DOI: https://doi.org/10.7554/eLife.36696.004

The following source data and figure supplement are available for figure 2:

**Source data 1.** The source data to plot the bar chart in *Figure 2A*.
DOI: https://doi.org/10.7554/eLife.36696.006
**Source data 2.** The source data to plot the bar chart in *Figure 2F*.
DOI: https://doi.org/10.7554/eLife.36696.007

*Figure 2 continued on next page*

*Figure 2 continued*

**Figure supplement 1.** A large number of Ezh2 peaks uniques in H3.3K27M mutant mouse ES cells is identified.

DOI: https://doi.org/10.7554/eLife.36696.005

Ezh2 peaks, and found that the H3.3K27M reads density was significantly higher at Ezh2 peaks unique in H3.3K27M mutant ES cells than those found in both wild type and mutant ES cells (*Figure 2C*). Interestingly, similar analysis on the published dataset using HA- and EYFP-tagged H3.3 proteins in mouse ES cells showed that wild type H3.3 was enriched to a similar level between these two groups of Ezh2 peaks (*Figure 2—figure supplement 1C*). The difference in the enrichment of H3.3K27M and wild type H3.3 at these two groups of Ezh2 peaks is currently unknown. Nonetheless, compared to wild type cells, H3K27me3 levels at the Ezh2 peaks unique to H3.3K27M mutant cells were dramatically reduced in the H3.3K27M mutant cells (*Figure 2D*). These results strongly support the idea that Ezh2 is redistributed to new sites in H3.3K27M mutant cells, but the presence of Ezh2 at the new sites does not generate H3K27me3, likely due to the presence of H3.3K27M mutant proteins at these sites (see below).

To analyze the properties of Ezh2 peaks found in H3.3K27M mutant cells, we then analyzed the overlap of these two groups of Ezh2 peaks with Ezh2, H3K4me3, H3K27me3, EP300, H3K27ac and H3K4me1 ChIP-seq peaks as well as annotated promoters, CpG island and bivalent promoter elements in wild type mouse ES cells from ENCODE database (*Figure 2E*). The common Ezh2 ChIP-seq peaks overlapped with high levels of H3K27me3 and were enriched with promoters including bivalent promoters and CpG island in wild type ES cells. These results suggest that these common Ezh2 peaks are classical and strong PRC2 sites, and that Ezh2 is recruited to these sites to retain H3K27me3 in H3.3K27M mutant ES cells. Surprisingly, the Ezh2 peaks unique to H3.3K27M mutant ES cells were overlapped with H3K27me3 and H3K4me1 in wild type ES cells. Chromatin regions containing both H3K4me1 and H3K27me3 are classified as poised enhancers (*Creyghton et al., 2010*).

To validate the ChIP-Seq results, we analyzed the enrichments of Ezh2 and H3.3K27M at three poised enhancers using ChIP-PCR. Ezh2 was indeed enriched at all three poised enhancers in H3.3K27M mutant ES cells compared to wild type ES cells. In contrast, Ezh2 was reduced at the Hoxa1 promoter, which contained high levels of H3K27me3, in H3.3K27M mutant ES cells compared to wild type ES cells (*Figure 2F*). H3.3K27M mutant proteins were enriched at these three poised enhancers compared to the Hoxa1 promoter, but to a significantly reduced degree compared to the actively transcribed gene, Actin, in H3.3K27M mutant ES cells (*Figure 2G*). These results strongly support the idea that Ezh2, possibly the PRC2 complex, is redistributed to new sites (poised enhancers) in H3.3K27M mutant cells, which leads to the reduction of the PRC2 complex and thereby H3K27me3 at the strong PRC2 sites.

In summary, most H3.3K27M mutant proteins, like wild type H3.3, are enriched at actively transcribed genes where the PRC2 complexes are absent. However, Ezh2, likely other subunits of PRC2 complex, is trapped at poised enhancers where H3.3K27M mutant proteins are detected, albeit at reduced levels compared to actively transcribed genes. This provides an explanation for our previous observation that more Ezh2 proteins were co-purified with H3.3K27M- than wild type H3.3-mononucleosomes (*Chan et al., 2013a*).

## A subgroup of Ezh2 ChIP-seq peaks co-localizes with H3.3K27M mutant proteins in SF8628 cell line

The absence of matched control of DIPG lines precluded us from finding whether Ezh2 is also redistributed to poised enhancer using the same method utilized in mouse ES cells. Therefore, we used 16,656 Ezh2 ChIP-seq peaks identified in DIPG line, SF8628, and performed unsupervised clustering analysis on their overlaps with H3.3K27M and H3K27me3 peaks in DIPG cells as well as their overlap with H3K4me1, H3K4me3 and H3K27me3 and annotated promoters in human neuro precursor cells (NPCs). We observed that Ezh2 peaks were separated into five different clusters (*Figure 2H*). Cluster 1 and 2 Ezh2 peaks overlapped with H3K27me3 and low levels of H3.3K27M mutant proteins in SF8628 cells. Ezh2 peaks in Cluster 1 and 2 also overlapped with promoters and H3K27me3 in human NPCs, consistent with the idea that these Ezh2 and H3K27me3 peaks likely represent Ezh2

peaks commonly found in tumor-initiating cells before the introduction of H3.3K27M mutation. Interestingly, the other three clusters of Ezh2 peaks overlapped with H3.3K27M mutant proteins, but with very low levels of H3K27me3. Analysis of Ezh2 and H3.3K27M mutant protein peaks in SF8628 cells using the published datasets (*Piunti et al., 2017*) also identified a group of Ezh2 peaks that co-localize with H3.3K27M mutant proteins, but with very low levels of H3K27me3 (*Figure 2—figure supplement 1D*). The Ezh2 peaks that co-localized with H3.3K27M mutant proteins were at regions marked by histone marks for weak promoters and/or poised enhancers in the reference human NPCs (*Figure 2G* and *Figure 2—figure supplement 1D*). Taken together, these results support the idea that the PRC2 complex is likely also redistributed to poised enhancers in DIPG cells, providing a mechanistic explanation for the global reduction of H3K27 methylation in cells expressing H3.3K27M mutant proteins.

## A fraction of H3K27me3 ChIP-seq peaks is conserved among DIPG patient tissue and two DIPG lines with the H3.3K27M mutation

In addition to the global reduction of H3K27 methylation, we and others paradoxically observed the enrichment of H3K27me3 within the promoters of hundreds of genes in the DIPG cell line (SF7761) harboring the H3.3K27M mutation (*Bender et al., 2013*; *Chan et al., 2013a*; *Lewis et al., 2013*). To determine whether the genes associated with locus-specific H3K27me3 are common in other DIPG tumor lines and in primary DIPG tissues, we analyzed H3k27me3 ChIP-seq in another DIPG xenograft cell line (SF8628), and one DIPG tissue sample from a surgical specimen with H3.3K27M mutation. The presence of H3K27M mutant proteins in the DIPG tissue was confirmed through Western blot (*Figure 3—figure supplement 1A*). We identified 2,356 H3K27me3 ChIP-seq peaks in SF7761, 5080 peaks in SF8628 and 12,033 peaks in DIPG patient tissue. When normalized against spike-in chromatin, the majority of these peaks exhibited reduced H3K27me3 levels compared to the reference human NSCs or reference brain tissue from a gliosis patient (*Figure 3—figure supplement 1B* and *Supplementary file 1*). About 9% of H3K27me3 ChIP-seq peaks identified in SF7761 or SF8628, and 6% of H3K27me3 peaks in primary DIPG tissue exhibited an increased level of H3K27me3 compared to the reference cells or reference brain tissue, respectively (*Figure 3—figure supplement 1C*). Importantly, of all the promoters with H3K27me3 ChIP-seq peaks, 676 promoters overlapped among these three samples (*Figure 3A and B*), which also overlapped with the ones associated with increased H3K27me3 peaks identified in the H3.3K27M transformed human NPCs (*Funato et al., 2014*) (*Figure 3—figure supplement 1D*). Finally, H3K27me3 surrounding the transcription start site (TSS) exhibited a similar pattern among the DIPG cell lines and the primary DIPG tissue (*Figure 3—figure supplement 1E and F*). These results indicate that a fraction of H3K27me3 peaks is likely conserved among DIPG samples with the H3.3K27M mutation.

## Low levels/absence of H3.3K27M mutant proteins at PRC2 sites helps retain H3K27me3 peaks in mouse ES cells and in DIPG cells with the H3.3K27M mutation

To understand how H3K27me3 peaks are retained in the environment of the global reduction of H3K27 methylation in cells with H3.3K27M mutant proteins, we analyzed the properties of H3K27me3 peaks in mutant mouse ES cells because of the presence of matched control of wild type ES cells. To do this, we first divided H3K27me3 peaks into two groups, most reduced and least reduced, based on the average levels of H3K27me3 in both H3.3K27M mutant mouse ES clones (*Figure 3C*). We then compared the average H3.3K27M ChIP-seq reads density between these two groups of gene promoters in H3.3K27M mutant cells. The gene promoters with the least reduced H3K27me3 had lower levels of H3.3K27M mutant proteins than those with most reduced H3k27me3 in mutant cells (*Figure 3D*). Moreover, we also compared the H3K27me3 ChIP-seq read density at the promoters of these two groups of genes in wild type ES cells (*Figure 3E*) and found that gene promoters with the least reduced H3K27me3 in H3.3 K27M mutant cells had higher levels of H3K27me3 in wild type cells than the gene promoters with the most reduced H3K27me3. Together, these results support the idea that the presence of H3K27me3 peaks in H3.3K27M mutant cells is likely due to two non-exclusive possibilities, high levels of H3K27me3 at these sites before the introduction of H3.3K27M mutant proteins and low levels of H3.3K27M mutant proteins assembled to these sites after expression of H3.3K27M mutant proteins.

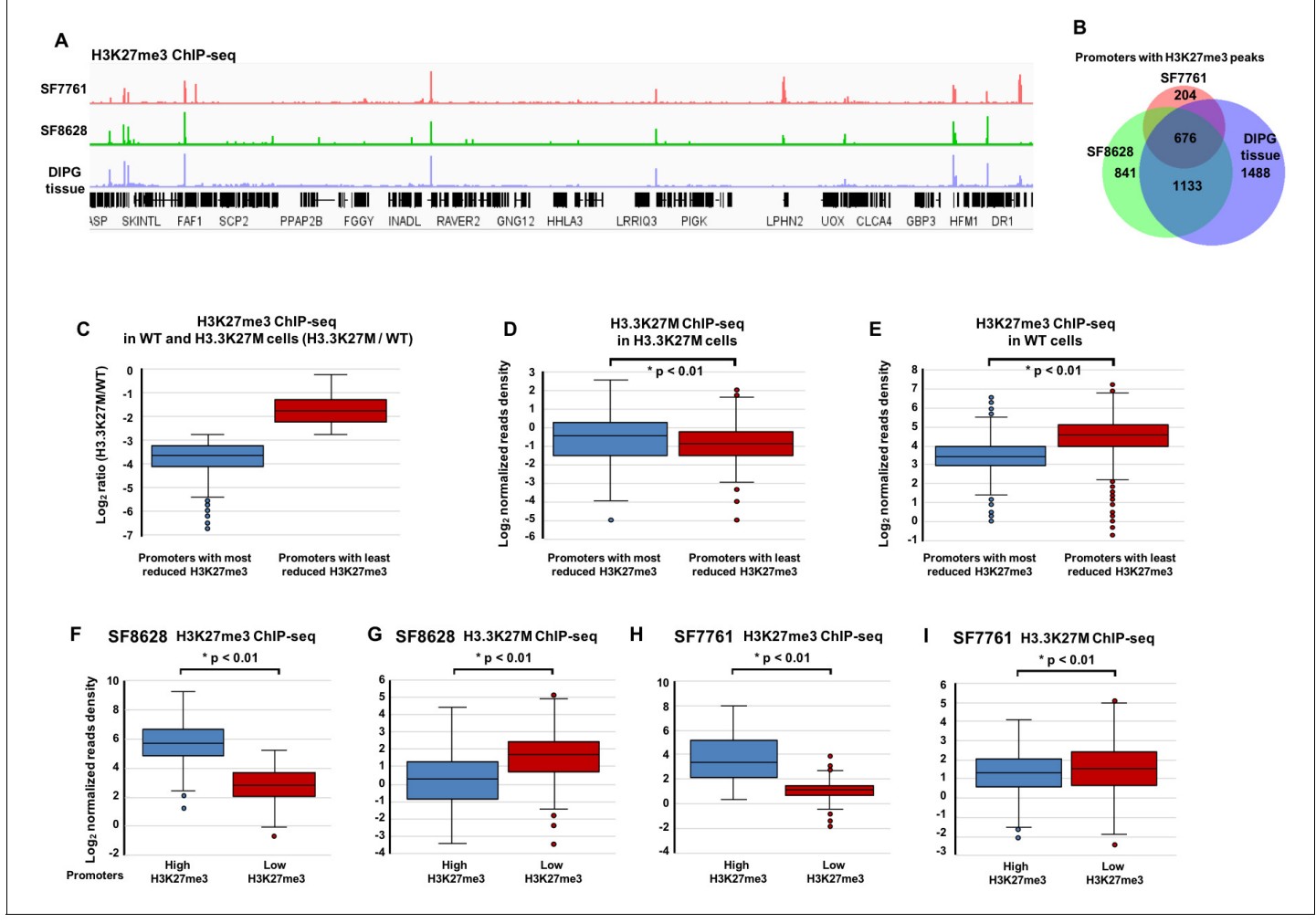

**Figure 3.** H3.3K27M levels on chromatin are low at gene promoters with H3K27me3. (A) Representative Integrative Genomics Viewer shows the distribution of H3K27me3 in two DIPG cancer cell lines (SF7761 and SF8628) and one DIPG primary tissue. RefSeq genes are shown at the bottom. (B) Venn diagram illustration represents the promoters with H3K27me3 peaks among SF7761, SF8628, and DIPG tissue. (C–E) Boxplot showing that H3.3K27M mutant proteins at gene promoters with the least reduced H3K27me3 are low compared to those with the most reduced H3K27me3 in H3.3K27M mutant mouse ES cells. Base on the average ratio of H3K27me3 ChIP-seq read density in H3.3K27M cells (C), H3K27me3 peaks in H3.3K27M mutant cells were separated into two groups, the most and least reduced, used to calculate the average read density of H3.3K27M ChIP-seq (D), and the read density of H3K27me3 ChIP-seq in WT cells (E). P value was calculated by two-tailed Student's t test. (F–I) The levels of H3K27me3 and H3.3K27M show an inverse-relationship in DIPG cells. 4195 and 3568 promoters with H3K27me3 ChIP-seq peaks were identified in 'tumour-initiating cells' for SF8628 and SF7761, respectively, before introducing the H3.3K27M mutation (see *Figure 3—figure supplement 2* for details). Based on the average H3K27me3 ChIP-seq read density in either SF8628 or SF7761 line, these promoters were separated into two groups, promoters with the highest H3K27me3 and promoters with lowest H3K27me3 in each line and used to calculate H3K27me3 read density and H3.3K27M read density in each line independently. P value was calculated by two-tailed Student's t test.

DOI: https://doi.org/10.7554/eLife.36696.008

The following figure supplements are available for figure 3:

**Figure supplement 1.** A small fraction of H3K27me3 peaks shows an equal or higher level of H3K27me3 in DIPG tumor cells than the corresponding reference cells.

DOI: https://doi.org/10.7554/eLife.36696.009

**Figure supplement 2.** In silico identification of H3K27me3 peaks at tumor-initiating cells of SF8628 and SF7761.

DOI: https://doi.org/10.7554/eLife.36696.010

Next, we analyzed whether similar observations hold true for the observed H3K27me3 peaks in DIPG tumor line, SF8628 and SF7761. Because vast difference in the number of H3K27me3 peaks was identified between these two samples, we analyzed H3K27me3 and H3.3K27M datasets from each line independently. To do this, we need to identify the PRC2 sites in 'tumor-initiating cells' of each DIPG line before the introduction of H3.3K27M mutant proteins. Because the expression of H3.3K27M in human NPCs resulted in neoplastic transformation (*Funato et al., 2014*), and mouse NSCs are proposed to be the cell origin of DIPG (*Monje et al., 2011*), we hypothesize that human NSCs or NPCs are 'tumor-initiating cells' of DIPG. We used two H3K27me3 ChIP-seq datasets, one from human NSCs that we generated (*Chan et al., 2013a*), and one from human NPCs downloaded from GEO (GSM896165) and identified 3212 overlapped H3K27me3 ChIP-seq peaks at promoters (*Figure 3—figure supplement 2A*). Because NPCs and NCSs are not matched controls for DIPG tumor-initiating cells, we also included H3K27me3 ChIP-seq peaks unique in SF8628 or SF7761 compared to NSCs or NPCs for the analysis. In total, 4,195 H3K27me3 peaks were identified and assumed to be present in 'tumor-initiating cells' of SF8628 before the introduction of H3.3K27M mutant proteins. We then divided the 4195 H3K27me3 promoters into two groups, high and low, based on the average levels of H3K27me3 at these gene promoters in SF8628 cell line, and compared the levels of H3.3K27M mutant proteins between these two groups of gene promoters (*Figure 3F*). We observed that the H3.3K27M mutant proteins were lower at the gene promoters with high levels of H3K27me3 than those with low levels of H3K27me3 (*Figure 3G*). Analysis of the H3K27me3 and H3.3K27M datasets obtained in SF7761 supported the same conclusion (*Figure 3H and I* and *Figure 3—figure supplement 2B*). These results suggest that the absence or low levels of H3.3K27M mutant proteins at the promoters with H3K27me3 help retain H3K27me3 at these genes, providing an explanation for the retention of H3K27me3 in cells expressing H3.3K27M mutant proteins.

## Many putative TSGs are silenced in DIPG cells likely through the H3K27me3-mediated mechanism

Silencing of tumor suppressor gene (TSG) p16 in DIPG cells by H3K27me3 and the PRC2 is important for tumorigenesis (*Cordero et al., 2017*; *Mohammad et al., 2017*; *Piunti et al., 2017*). Because we have identified 676 genes with H3K27me3 peaks in three samples (SF8628, SF7761 and the primary DIPG tissue), we asked whether any of these genes are TSGs. Based on the annotation of three different databases, we found that 80 of the 676 genes are TSGs (*Figure 4—figure supplement 1A*). To identify the candidate TSGs whose silencing is important for the proliferation of DIPG cells, we focused on the analysis of 12 putative TSGs including p16 that showed the lowest expression in SF7761 and SF8628 cells based on RNA-seq datasets we collected (*Chan et al., 2013a*). First, we tested their expression levels in 5 DIPG lines (SF8628, SF7761, DIPG17, DIPG13, and PED8) with H3.3K27M mutation, one DIPG line with H3.1K27M mutation (DIPG4), one brain tumor line with H3.3G34V mutation (KNS42) and one high grade glioma cell line with wild type H3.3 (SF9427) by RT-PCR. While the expression of these 12 genes in all these tumor lines was low in general, clustering analysis showed that the expressions of NGFR, WT1 and MME were lowest among these 12 genes in these tumor cells, irrespectively mutation status of histone H3, when compared with human NSCs (*Figure 4A*). Moreover, the involvement of these genes in DIPG tumors has not been reported before. Therefore, we focused our analysis on the three genes with lowest expression levels.

Next, we analyzed the enrichment of H3K27me3 and Ezh2 at the promoters of NGFR, WT1 and MME in these cell lines and human NSCs, and observed that H3K27me3 and Ezh2 were enriched at the promoters of WT1 promoter more than other two gene promoters in all these tumor lines tested (*Figure 4—figure supplement 1B and C*). We also evaluated the effect of Suz12 depletions on the expression of these three genes as well as p16 in two DIPG lines (SF7761 and SF8628). Depletion of Suz12 using two independent shRNAs resulted in increased expression of all these four genes in two DIPG lines tested (*Figure 4B*). Together, these results indicate that in addition to p16, the PRC2 complex is involved in silencing of multiple TSGs in DIPG tumor cells.

Finally, we analyzed the expression of p16, NGFR, WT1 and MME in 37 DIPG patient samples with H3.3K27M mutation and 58 non-brainstem pediatric high-grade glioma (NBS-HGG) samples with wild type H3.3 using the public database (*Wu et al., 2014*). The expression of WT1 and p16 was significantly lower in DIPG tumor samples than NBS-HGG samples (*Figure 4C*). In contrast, the expressions of NGFR or MME were similar between DIPG and NBS-HGG samples (*Figure 4—figure*

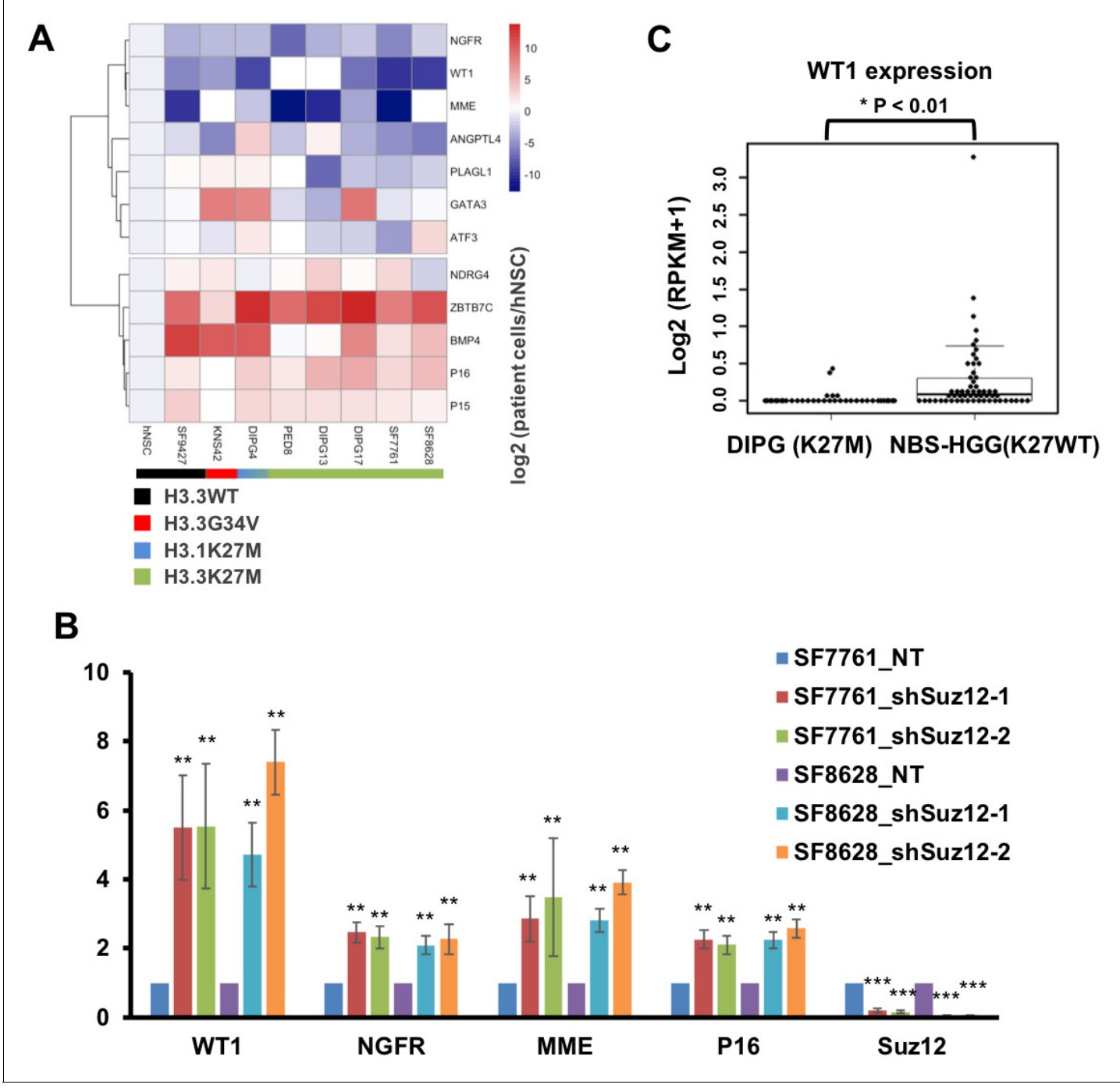

**Figure 4.** Several TSGs including WT1 are silenced through H3K27me3-mediated mechanism in DIPG cells with H3.3K27M mutation. (A) Analysis of expression levels of 12 different tumor suppressor genes (TSGs) in different DIPG lines and human NSC. Expressions of 12 different TSGs in tumor lines, with the mutation status of histone H3 shown at the bottom, were analyzed by quantitative RT-PCR and clustered hierarchically. Data are mean (N = 3 independent replicates). (B) Depletion of Suz12 increases the expression of WT1 in DIPG lines. The expression levels of three most silenced TSGs as well as p16 were analyzed in two H3.3K27M mutant tumor cells (SF7761 and SF8628) after depletion of Suz12. Data are mean ± SD (N = 3 independent replicates, **p<0.01). (C) Box plot of gene expression levels of WT1 in DIPG with H3.3K27M (n = 37) and non-brainstem pediatric high-grade glioma (NBS-HGG) with wild type H3.3 (n = 58) tumor tissues (*Wu et al., 2014*). P value was calculated by two-tailed Student's t test.

DOI: https://doi.org/10.7554/eLife.36696.011

The following source data and figure supplement are available for figure 4:

**Source data 1.** The source data to plot the bar chart in *Figure 4B*.

DOI: https://doi.org/10.7554/eLife.36696.013

*Figure 4 continued on next page*

*Figure 4 continued*

**Figure supplement 1.** H3K27me3 and Ezh2 are enriched at the promoters of TSGs in DIPG cells.
DOI: https://doi.org/10.7554/eLife.36696.012

*supplement 1D*). These results indicate that in addition to p16, WT1, a TSG first discovered in the pediatric kidney tumor, Wilms Tumor (*Gessler et al., 1990*; *Maiti et al., 2000*), is also silenced by the H3K27me3-mediated mechanisms in DIPG tumors.

## Forced expression of WT1 inhibits the proliferation of H3.3K27M mutant cells

Next, we asked whether forced expression of WT1 affected the proliferation of DIPG cells. We used two methods to increase WT1 expression. First, we overexpressed each of the four different iso-forms of WT1 in H3.3K27M mutant cells (SF7761 and SF8628) and H3.3 wild type cells (SF9427), and found that the expression of each WT1 isoform inhibited the proliferation of two H3.3K27M mutant cells, but not H3.3 wild type cells (SF9427) (*Figure 5—figure supplement 1A and B*), suggesting that reduced levels of WT1 are more important for the proliferation of these two DIPG lines than SF9427 cells. Second, we tested whether alterations in chromatin states of the WT1 promoter by tar-geting p300 using CRISPR/dCas9 (catalytic dead Cas9) would also change the expression of WT1 and cell proliferation. We targeted dCas9-HA and dCas9-HA fused with p300 catalytic domain (dCas9-HA-p300-HA) to the WT1 promoter in SF8628 cells using two independent sgRNAs (*Figure 5—figure supplement 1C*). The dCas9-HA and dCas9-HA-p300-HA were enriched at the WT1 promoter only when they were co-expressed with either of the two sgRNAs. Moreover, dCas9-HA and dCas9-HA-p300-HA were not at the promoter of CDK6 isoform NM_001145306 (*Figure 5A*). Importantly, targeting dCas9-HA-p300-HA, but not dCas9-HA alone to the WT1 promoter led to a reduction of H3K27me3 and Ezh2 as well as an increase of H3K27ac at the WT1 promoter (*Figure 5A*). Finally, the expression level of WT1 increased (*Figure 5B*) and the proliferation of SF8628 cells was inhibited (*Figure 5C*) when targeting dCas9-HA-p300-HA, but not dCas9-HA alone, to the WT1 promoter. Together, these results show that forced increase in H3K27ac at the WT1 pro-moter in SF8628 cells leads to reduced H3K27me3, increased expression of WT1 and decreased pro-liferation of DIPG cells, supporting the idea that H3K27me3-mediated silencing of WT1 supports the proliferation of DIPG cells.

## Discussion

It remains in a debate on how the expression of H3.3K27M mutant proteins leads to a global reduc-tion of H3K27me3 in DIPG cells as well as in any cell types expressing the H3.3K27M mutant protein. Using mouse ES cells knocked-in with the same H3.3K27M mutation found in most DIPG lines, we observed that Ezh2, most likely the PRC2 complex, is redistributed to poised enhancers containing both H3K4me1 and low levels of H3K27me3, but not at promoters or gene bodies of highly tran-scribed genes where H3.3K27M mutant proteins are high. In DIPG cells, we also observed that a group of Ezh2 peaks co-localizes with H3.3K27M mutant proteins, likely also at poised enhancers. These results support a model whereby some PRC2 molecules are sequestered at these gene regula-tory elements. Because the expression of key subunits of the PRC2 complex is not affected in H3.3K27M mutant cells, we suggest that the sequestration of PRC2 complex at these poised enhancers contributes to the global reduction of H3K27me3 in cells expressing H3.3K27M mutant proteins (*Figure 5D*). We also observed that H3.3K27M mutant proteins are low at gene promoters with high H3K27me3 in H3.3K27M mutant mouse ES cells as well as DIPG cells. These results indi-cate that low levels or absence of H3.3K27M mutant proteins contributes to the retention of H3K27me3 at these sites. Finally, we show that WT1 is silenced through the H3K27me3-mediated mechanism, and forced expression of WT1 inhibits the proliferation of DIPG cells. Together, these studies provide the mechanistic insight into reprograming H3K27me3 by H3.3K27M mutant proteins and identify a novel TSG that is silenced in DIPG tumor cells for the proliferation of tumor cells.

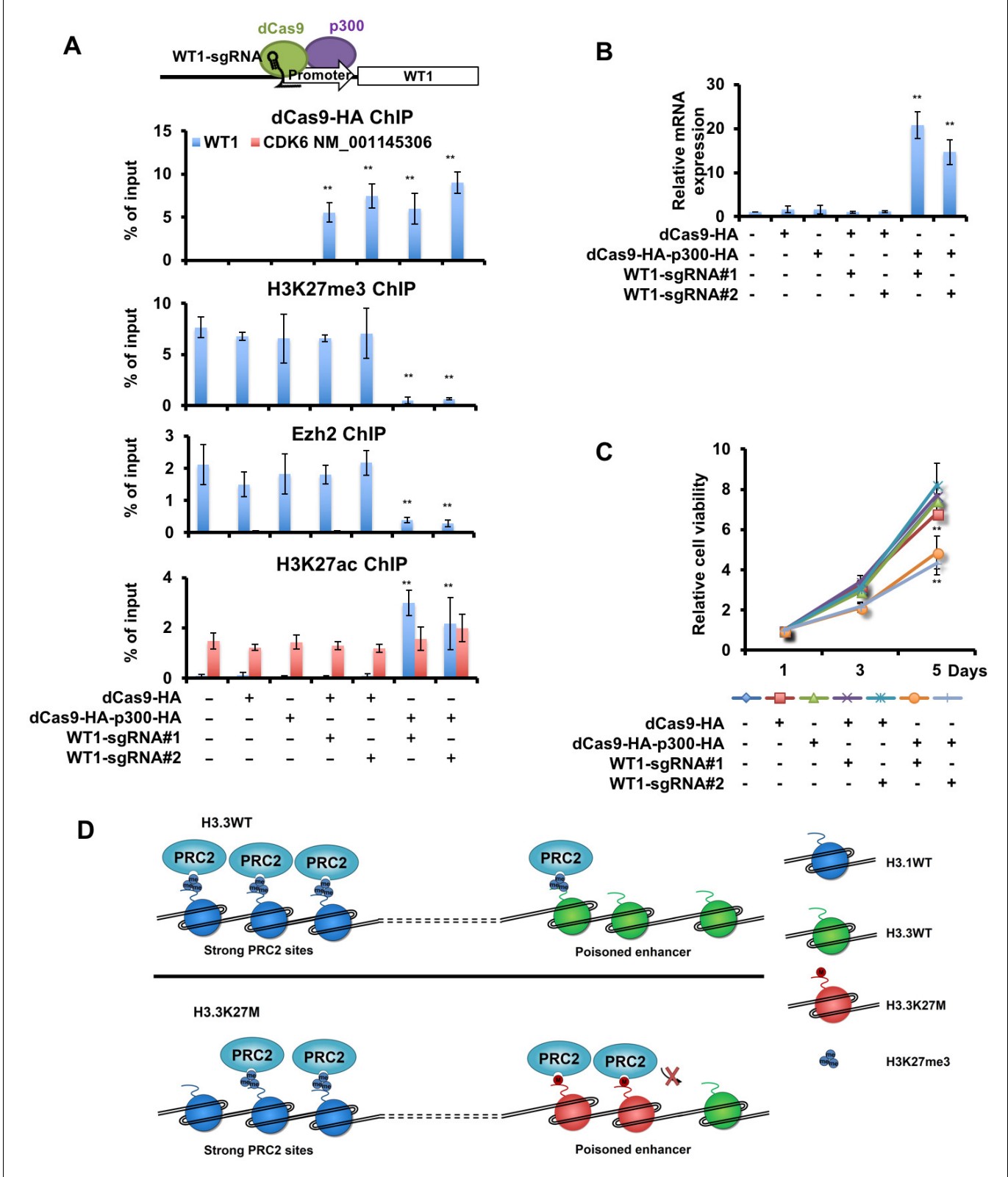

**Figure 5.** An increase in WT1 expression inhibits the proliferation of H3.3K27M mutant cells. (**A**) Targeting p300 to the WT1 promoter results in an increase in H3K27ac and a reduction of H3K27me3. Top: a schematic strategy for targeting dCas9-HA-p300-HA fusion proteins to the promoter of WT1. ChIP-PCR analysis of dCas9-HA and dCas9-HA-p300-HA fusion proteins (first panel), H3K27me3 (second panel), Ezh2 (third panel), and H3K27ac (last panel) at the promoter of WT1 was performed using SF8628 cells transfected with indicated combinations of dCas9-HA, dCas9-HA-p300-HA and two
*Figure 5 continued on next page*

*Figure 5 continued*

different sgRNAs targeting to the WT1 promoter. Data are mean ± SD (N = 3 independent replicates, **p<0.01). (B) The expression of WT1 increases after targeting dCas9-HA-p300-HA to the WT1 promoter. The expression of WT1 was analyzed by quantitative RT-PCR. Data are mean ± SD (N = 3 independent replicates, **p<0.01). (C) SF8628 cell proliferation is inhibited by targeting dCas9-HA-p300-HA to the WT1 promoter. Data are mean ± SD (N = 3 independent replicates, **p<0.01). (D) A model for reduction and retention of H3K27me3 in cells with H3.K27M mutation. In wild type cells, the PRC2 complex co-localizes with H3K27me3 peaks at the major PRC2 sites. Since these genes are silenced, the levels of H3.3 are low. Poised enhancers contain low levels of H3K27me3, PRC2 complex and H3.3. In H3.3K27M mutant cells, the PRC2 complexes, once recruited to the poised enhancers, are trapped there, likely due to higher concentration of H3.3K27M mutant proteins at poised enhancers. This will lead to a reduction of the PRC2 complex at strong PRC2 sites and a global reduction of H3K27me3. However, the amount of H3.3K27M mutant proteins at major PRC2 sites are low and thus having limited effect on the ability of the PRC2 complex, which is at reduced levels compared to wild type cells, to methylate H3K27.

DOI: https://doi.org/10.7554/eLife.36696.014

The following source data and figure supplement are available for figure 5:

**Source data 1.** The source data to plot the bar chart in *Figure 5A*.
DOI: https://doi.org/10.7554/eLife.36696.016
**Source data 2.** The source data to plot the bar chart in *Figure 5B*.
DOI: https://doi.org/10.7554/eLife.36696.017
**Source data 3.** The source data to plot the bar chart in *Figure 5C*.
DOI: https://doi.org/10.7554/eLife.36696.018
**Figure supplement 1.** Forced expression of WT1 inhibits the proliferation of DIPG cells with H3.3K27M mutation.
DOI: https://doi.org/10.7554/eLife.36696.015

## How are the PRC2 complexes redistributed to poised enhancers?

How H3.3K27M mutant proteins affect H3K27me3 levels in cells remain in a debate (*Chan et al., 2013a*; *Justin et al., 2016*; *Lewis et al., 2013*; *Piunti et al., 2017*). Using mouse ES cells knocked-in with the H3.3K27M mutation found in DIPG cells, we observed that H3.3K27M mutant proteins are enriched at actively transcribed genes where H3K27me3 and PRC2 are absent. In addition, Ezh2 proteins are detected at many more loci in H3.3K27M mutant than wild type ES cells. These Ezh2 peaks unique in H3.3K27M mutant ES cells are enriched with poised enhancers as well as H3.3K27M mutant proteins compared to those Ezh2 peaks commonly found in both wild type and H3.3K27M mutant ES cells. These results support the model that the PRC2 complex is redistributed to poised enhancers in H3.3K27M mutant ES cells. Because the total levels of the PRC2 complex are not altered in H3.3K27M mutant ES cells, the enrichment of the PRC2 complex at poised enhancers will lead to the reduced amounts of the PRC2 complex to methylate the PRC2 sites. While we do not have a perfect control for our analysis of DIPG cells, we also found that a group of Ezh2 peaks co-localize with H3.3K27M mutant proteins in DIPG line SF8628 based on analysis of Ezh2 and H3.3K27M ChIP-seq datasets from ours and published ones. These results indicate that the reduction of H3K27me3 in cells expressing H3.3K27M mutant proteins including DIPG cells is likely due, at least in part, to the redistribution of the PRC2 complex to poised enhancers.

How is the PRC2 complex redistributed to poised enhancers instead of active promoters with high levels of H3.3K27M mutant proteins? Two properties at the poised enhancers likely contribute to the retention of the PRC2 complex at these loci in H3.3K27M mutant cells. First, these poised enhancers, in general, contain low levels of H3K27me3 in wild type cells, suggesting that the PRC2 complex can be recruited to these loci, but cannot be detected at these loci in wild type cells because of its low affinity and/or low residence time at these sites. Moreover, it is known that H3.3 proteins localize at gene regulatory elements including poised enhancers in mouse ES cells (*Goldberg et al., 2010*), and we observed that H3.3K27M mutant proteins can be assembled to these poised enhancers, although at a lower level compared to actively transcribed genes. Therefore, we propose that the PRC2 complex is recruited to poised enhancers independent of H3.3K27M mutant proteins but stays at these sites much longer because of the presence of the H3.3K27M mutant proteins. We speculate that while the levels of H3.3K27M mutant proteins at the gene promoters of actively transcribed genes are high, the PRC2 complex cannot be recruited to these sites initially and therefore the PRC2 complex cannot be sequestered at these sites despite the presence of high levels of H3.3K27M mutant proteins. Thus, we propose that the ability of poised enhancers to recruit the PRC2 complex, as well as the presence of H3.3K27M mutant proteins at these gene regulatory elements, contributes to the sequestration of the PRC2 at these gene

regulatory elements in mouse ES cells and possibly in DIPG cells with the H3.3K27M mutation. In the future, it will be very interesting to test this model once we know how the PRC2 complex is recruited to poised enhancers.

While our manuscript is under revision, it has been observed that the residence time and search time of Ezh2 are prolonged in the H3.3K27M mutant ES cells using live-cell single-molecule imaging, while the total protein levels of Ezh2 on chromatin are not changed (*Tatavosian et al., 2018*). The prolonged search time of Ezh2 may either reflect the trapping of Ezh2 at the poised enhancers and/ or an alternative mechanism that also contributes to the reduction of H3K27me3.

About 20% DIPG tumors contain the mutation at genes encoding histone H3.1 (*Nikbakht et al., 2016*). H3.1 is assembled into nucleosomes through DNA replication-coupled nucleosome assembly. Moreover, H3.1 is predominantly localized at heterochromatin, which is in contrast to H3.3 localization at genic regions. Therefore, it would be interesting to determine how Ezh2 and PRC2 complex are distributed in cells expressing the H3.1K27M mutant protein in the future. In addition to H3.3K27M mutation, cells containing H3.3K36M mutation found in chondroblastoma exhibit a global reduction of H3K36 methylation (*Fang et al., 2016*; *Lu et al., 2016*). We have shown that H3.3K36M mutant proteins are enriched at gene bodies of actively transcribed genes. The present study also begs the questions on whether the H3K36 methyltransferases are also redistributed in H3.3K36M mutant cells and contribute to the reduction of H3K36 methylation.

## Mechanism for the retention of H3K27me3 in DIPG cells

We and others also observed that H3K27me3 peaks were present at hundreds of loci based on the analysis of H3K27me3 ChIP-seq (*Bender et al., 2013*; *Chan et al., 2013a*). In this study, we further confirm the dichotomous changes of H3K27me3 in one primary DIPG tumor sample. Importantly, we show that a significant number of gene promoters with H3K27me3 is common among three DIPG samples analyzed (*Figure 3B*). In H3.3K27M mutant mouse ES cells, almost all retained H3K27me3 ChIP-Seq peaks are also found in wild type mouse ES cells, but at a reduced level compared to wild type mouse ES cells. These results suggest that these H3K27me3 peaks in the mutant mouse ES cells are the bona fide PRC2 sites. H3.3K27M mutant proteins at H3K27me3-containing loci are either absent or at very low levels in both mouse ES cells and DIPG cells. In vitro, H3.3K27M mutant proteins inhibit the activity of the PRC2 complex. Similarly, replacement of other histone lysine residues to methionine such as H3K9 and H3K36 also inhibits the enzymatic activity of their corresponding lysine methyltransferase in vitro (*Fang et al., 2016*; *Herz et al., 2014*; *Jayaram et al., 2016*; *Lu et al., 2016*; *Shan et al., 2016*). In fission yeast, overexpression of Clr4, the H3K9 methyltransferase, overcomes the inhibition of H3K9M mutant proteins on H3K9me3 (*Shan et al., 2016*). Thus, the absence of or a low level of H3.3K27M mutant proteins at the gene promoters with H3K27me3 may be insufficient to inhibit the enzymatic activity of the PRC2 complex, leading to the retention of H3K27me3 at these sites (*Figure 5D*).

It is known that high levels of H3.3 are enriched at highly transcribed genes compared to lowly expressed genes (*Chow et al., 2005*; *McKittrick et al., 2004*; *Schübeler et al., 2004*; *Stroud et al., 2012*). H3.3K27M mutant proteins exhibit the same localization pattern in two DIPG lines as well as in H3.3K27M mutant mouse ES cells. We suggest that low levels of H3.3K27M mutant proteins detected at these gene promoters of strong PRC2 sites are likely due to the low expression level of these PRC2 regulated genes.

## Epigenetic silencing of WT1 is important for the proliferation of DIPG cells

Genetic studies using mouse models indicate that Ezh2 is required for both the initiation and maintenance of tumors (*Mohammad et al., 2017*), demonstrating the importance of the retention/gain H3K27me3 in tumorigenesis. Moreover, it has been shown that p16 is repressed through H3K27me3-mediated silencing (*Mohammad et al., 2017*; *Piunti et al., 2017*). In agreement with their studies, we also observed that depletion of Suz12 results in increased expression of p16. We extend this observation and show that WT1 is silenced through the H3K27me3-mediated mechanism and WT1 silencing is important for the proliferation of DIPG cells. Like p16, the expression of WT1 is low in DIPG tumor samples compared to NBS-HGG samples with wild type H3.3. WT1 is a zinc finger transcription factor (*Bardeesy and Pelletier, 1998*). It was first identified as a TSG in Wilms tumor, a

tumor also found exclusively in children (*Gessler et al., 1990*; *Maiti et al., 2000*). Accumulating evidence suggests that WT1 is also expressed in many different classes of intracranial tumors, including gliomas (*Izumoto et al., 2008*), oligodendrogliomas (*Rauscher et al., 2014*), ependymomas (*Yeung et al., 2013*), and meningiomas (*Iwami et al., 2013*) and functions as an oncogene in these tumors. We suggest that H3K27me3 and the PRC2 likely silence multiple TSGs including p16 and WT1 for the initiation and maintenance of DIPG tumors.

About 6 to 9% H3K27me3 peaks had a higher level of H3K27me3 in DIPG lines and tissues than reference cells or tissue, respectively. However, almost all retained H3K27me3 peaks in H3.3K27M mutant mouse ES cells exhibit reduced levels compared to wild type mouse ES cells. The apparent discordance between these two results suggests that the reference cells (NPCs and NSCs) or tissues used as controls for DIPG cells and tissues are not perfect. Alternatively, some of these high levels of H3K27me3 peaks detected in DIPG cells are selected for the 'tumor-initiating cells' to gain the proliferative advantage. Supporting this idea, high levels of H3K27me3 are detected at the down-regulated genes including NKX2.2 gene in H3.3K27M transformed NPCs compared to control cells (*Funato et al., 2014*). It is proposed that this will facilitate the resetting of neuro-precursor into stem cells state. Moreover, TSGs such as WT1 are likely expressed in tumor-initiating cells, and consequently, contain low levels of H3K27me3 at their gene promoters. Therefore, we speculate that selective pressure will lead to an increase of H3K27me3 at these gene promoters, which leads to silencing of these TSGs and drives tumorigenesis in DIPG cells.

The dichotomous change in H3K27me3 in DIPG cells is reminiscent of DNA methylation alteration observed in many cancer cells (*Baylin and Jones, 2011*). Genomic DNA in most cancer cells is hypomethylated (*Eden et al., 2003*; *Ehrlich, 2009*), and yet DNA hypermethylation at the promoters of TSGs has been frequently reported in many cancer types (*Baylin and Jones, 2011*; *Esteller, 2007*; *Struhl, 2014*). DNA methylation analysis in DIPG cells indicates that DNA is hypomethylated (*Bender et al., 2013*). Therefore, we propose that it is the dichotomous changes in H3K27me3, including both the global loss and locus retention/gain of H3K27me3, that contribute to tumor initiation and maintenance of DIPG.

# Materials and methods

## Key resources table

| Reagent type (species) or resource | Designation | Source or reference | Identifiers | Additional information |
|---|---|---|---|---|
| Cell line (*H. sapiens*) | SF7761 | PMID: 23603901 | RRID:CVCL_IT45 | Growth in neural stem cell media as neurosphere |
| Cell line (*H. sapiens*) | SF8628 | PMID: 25401693 | RRID:CVCL_IT46 | Growth in DMEM as monolayer |
| Cell line (*H. sapiens*) | KNS42 | PMID: 25401693 | RRID:CVCL_0378 | Growth in DMEM as monolayer |
| Cell line (*H. sapiens*) | SF9427 | PMID: 25401693 | | Growth in DMEM as monolayer |
| Cell line (*H. sapiens*) | PED8 | PMID: 26376656 | | Growth in neural stem cell media as neurosphere |
| Antibody | H3K27me3 | Cell Signaling | C36B11, RRID:AB_2616019 | 1.4 µg for ChIP, 1:1000 for Western blot |
| Antibody | Ezh2 | Cell Signaling | 5246, RRID:AB_10694683 | 5 µg for ChIP, 1:1000 for Western blot |
| Antibody | Suz12 | Cell Signaling | 3737S, RRID:AB_2196850 | 1:1000 for Western blot |
| Antibody | H3K27ac | Abcam | ab4729, RRID:AB_2118291 | 2 µg for ChIP, 1:1000 for Western blot |

*Continued on next page*

*Continued*

| Reagent type (species) or resource | Designation | Source or reference | Identifiers | Additional information |
|---|---|---|---|---|
| Antibody | H3K4me1 | Abcam | ab8895, RRID:AB_306847 | 2 μg for ChIP, 1:1000 for Western blot |
| Antibody | α-Tubulin | Sigma | T9026, RRID:AB_477593 | 1:1000 for Western blot |
| Antibody | FLAG | Sigma | F1804, RRID:AB_262044 | 2 μg for ChIP, 1:1000 for Western blot |
| Antibody | H3K27M | Millipore | ABE419, RRID:AB_2728728 | 2 μg for ChIP, 1:1000 for Western blot |
| Antibody | HA | This study | 12CA5 | 2 μg for ChIP, 1:1000 for Western blot |
| Antibody | H3 | PMID: 23603901 | | 1:10000 for Western blot |
| Commercial assay or kit | SuperScript III First-Strand kit | Invitrogen | 18080–051 | |
| Commercial assay or kit | RNeasy Mini Kit | Qiagen | 74104 | |
| Commercial assay or kit | QiaQuick PCR purification kit | Qiagen | 28104 | |
| Commercial assay or kit | ssoAdvanced Universal SYBR Green Supermix | Bio-Rad | 1725274 | |
| Commercial assay or kit | cell titer blue assay kit | Promega | G8081 | |
| Software, algorithm | Bowtie2 | PMID: 22388286 | RRID:SCR_005476 | |
| Software, algorithm | MACS2 | PMID: 18798982 | RRID:SCR_013291 | |
| Software, algorithm | Bedtools | PMID: 20110278 | RRID:SCR_006646 | |

## Cell lines

The patient-derived cell lines SF7761 and SF8628 harboring the histone H3.3 K27M mutation were used as described before. Briefly, SF7761 cells were cultured in human neural stem cells media and formed neurosphere. SF8628 cells were cultured as adherent monolayer cells in DMEM media (*Chan et al., 2013a*). KNS42 and SF9427 were cultured as previously described (*Hashizume et al., 2014*). DIPG4, DIPG13, and DIPG17 were kind gifts from Dr. Michelle Monje (Stanford University School of Medicine). PED8 was generated at Mayo Clinic by Dr. David Daniels as described (*Zhang et al., 2016*). Human neural stem cells (Cat.# N7800-100) were purchased from Invitrogen. HEK293T cells were purchased from ATCC.

Cell lines with H3.3K27M mutation used in this study were identified by PCR and sequencing. Mouse ES cells were analyzed with stemness by testing the expression of OCT4, Sox2, and Nanog, as well as alkaline phosphatase staining. In addition, mouse ES cells were analyzed with karyotyping and showed normal karyotype. All cell lines were tested negative for mycoplasma using a qPCR-based testing with mixed primers: 5'-cgcctgagtagtacgttcgc-3', 5'-cgcctgagtagtacgtacgc-3', 5'-tgcctgggtagtacattcgc-3', 5'-tgcctgagtagtacattcgc-3', 5'-cgcctgagtagtatgctcgc-3', 5'-cgcctgggtagta-cattcgc-3', 5'-gcggtgtgtacaagacccga-3', 5'-gcggtgtgtacaaaacccga-3', 5'-gcggtgtgtacaaaccccga-3'. None of the cell lines used in this study were included in the list of commonly misidentified cell lines maintained by International Cell Line Authentication Committee.

## Antibodies

Antibodies against H3K27me3 (Cat.# 9733), Ezh2 (Cat.# 5246), and Suz12 (Cat.# 3737) were purchased from Cell Signaling. Antibodies against H3K27ac (Cat.# ab4729) and H3K4me1 (Cat.# ab8895) were purchased from Abcam. Antibodies against α-Tubulin (Cat.# T9026) and FLAG (Cat.#

F1804) were purchased from Sigma. Antibody against H3K27M (Cat.# ABE419) was purchased from Millipore. Antibodies against the HA epitope (Clone No. 12CA5) and histone H3 were previously described (*Chan et al., 2013a*; *Han et al., 2013*).

## Oligonucleotides used

See *Supplementary file 1* for oligonucleotides used.

## Cell proliferation assays

$3 \times 10^3$ cells were plated in each well of 96-well plate with 100 µL of growth medium. Relative amounts of cells at the time after seeding were measured by the cell titer blue assay kit (Promega, Cat.# G8081).

## Reverse transcription (RT)-PCR

Total RNA was extracted using the RNeasy plus kit (Qiagen, Valencia, CA). cDNAs were synthesized using 0.5 µg of total RNA, random hexamers and Superscript III Reverse Transcriptase. Real-time PCRs were performed with 0.1 µM primers and SYBR Green PCR Master Mix (Bio-Rad). β-actin was used as a control to normalize the expression of target genes.

## Chromatin immunoprecipitation-deep sequencing (ChIP-seq) and ChIP-PCR

To prepare ChIP-seq samples for frozen tissues, samples were cut into 50 mg aliquots and homogenized on ice for 30 s in 500 µl 1X PBS. Cells or tissue homogenates were crosslinked with 1% formaldehyde for 10 mins and then quenched with 125 mM glycine for five mins at room temperature. Samples were washed with cold TBS twice before resuspended in cell lysis buffer (10 mM Tris-HCl, pH7.5, 10 mM NaCl, 0.5% NP-40) and incubated on ice for 10 mins. Lysates were washed and resuspended in 500 µl Mnase digestion buffer (20 mM Tris-HCl, pH 7.5, 15 mM NaCl, 60 mM KCl, 1 mM $CaCl_2$) in the presence of 1000 units of Mnase (NEB, Cat.# M0247S). After 20 mins incubation at 37°C with continuous mixing, digestion was stopped with 500 µl of sonication buffer (100 mM Tris-HCl, pH8.1, 20 mM EDTA, 200 mM NaCl, 2% Triton X-100, 0.2% sodium deoxycholate). Samples were sonicated for 15 mins (30 secs on/30 secs off). The chromatin content was estimated by the Qubit assay. For the normalization of ChIP efficiency, the chromatin prepared similarly from S2 or Sf9 cells was added to 1 to 5% of total chromatin. The chromatin was then incubated with 5 µg of rabbit monoclonal anti-Ezh2 antibody (Cell Signaling, Cat.# 5246), 1.4 µg of rabbit monoclonal anti-H3K27me3 antibody (Cell Signaling, Cat.# 9733), 2 µg of rabbit polyclonal anti-H3K27ac antibody (Abcam, Cat.# ab4729), 2 µg of rabbit polyclonal anti-H3K27M antibody (Millipore, Cat.# abe419), 2 µg of mouse monoclonal anti-Flag antibody (Sigma, Cat.# F1804), 2 µg of mouse monoclonal anti-HA antibody (Clone No. 12CA5), or 2 µg of rabbit polyclonal anti-H3K4me1 antibody (Abcam, Cat.# ab8895) on a rocker overnight. 30 µl of protein G-magnetic beads were added for additional 3 hr incubation. The beads were washed extensively with ChIP buffer (50 mM Tris-HCl, pH8.1, 10 mM EDTA, 100 mM NaCl, 1% Triton X-100, 0.1% sodium deoxycholate), high salt buffer (50 mM Tris-HCl, pH8.1, 10 mM EDTA, 500 mM NaCl, 1% Triton X-100, 0.1% sodium deoxycholate), $LiCl_2$ buffer (10 mM Tris-HCl, pH8.0, 0.25 M $LiCl_2$, 0.5% NP-40, 0.5% sodium deoxycholate, 1 mM EDTA), and TE buffer. Bound DNA was eluted and reverse-crosslinked at 65°C overnight. After the treatment of RNase A and proteinase K, DNAs were purified using Min-Elute PCR purification kit (Qiagen). 10 ng ChIP and input DNA were processed for library preparation by following the Ovation ultralow DR Multiplex kit (NuGEN). The ChIP-seq library DNA was sequenced using 51 bp pair-end sequencing on an Illumina HiSeq 2000/2500 instrument at the Center for Individualized Medicine Medical Genomics Facility, Mayo Clinic.

## ChIP-seq data analysis

Paired-end sequencing reads from H3K27me3, Ezh2, and H3.3K27M ChIP-seq were aligned to the human genome (hg19) or mouse genome (mm9) using the Bowtie2 software (*Langmead and Salzberg, 2012*). The consistent pair reads were used for the further analysis. After removal of PCR duplication reads by SAMtools (*Li et al., 2009*), we used BEDTools (*Quinlan and Hall, 2010*) and in-house Perl programs to calculate the genome-wide reads coverage. ChIP-seq peaks were identified by MACS2 (*Feng et al., 2012*) with the parameter of broad peak calling and the cutoff p-value set to $1 \times$

$10^{-5}$. The reads density scan was performed by in-house Perl programs using the normalized method: Reads Per Kilobase per 10 Million mapped reads. In order to calculate the average profile of H3.3K27M ChIP-seq, the gene annotation by UCSC was split into three groups (high expression genes, medium express genes, and low express genes) based on the expression levels of RNA-seq. Then the average read coverage for these three groups of genes was calculated across the annotated gene regions. Coverage across the annotated region which is the transcription start sites (TSS) to transcription end sites (TES) of each gene was calculated in 100 equally spaced bins. In addition, coverage was calculated in 100 equally spaced bins for both up-stream and down-stream flanking regions. The promoters used in this study were defined as 2 kb upstream and 500 bp downstream of TSS (base on UCSC annotation genes transcription start sites). Because we are using the MNase-digested chromatin to perform the ChIP-seq, it may contribute to high enrichment of H3.3K27M at TSS and TES.

## Spike-in chromatin H3K27me3 ChIP-seq analysis

For H3K27me3 ChIP-seq using the patient tissue, chromatin from sf9 cells was used as the spike-in reference. For H3K27me3 ChIP-seq in SF7761, SF8628 and mouse ES cells, chromatin from S2 cells was used as the spike-in reference. The sequence reads from H3K27me3 ChIP-seq were mapped and normalized according to published procedures (*Orlando et al., 2014*). Briefly, a combined genome sequence from human (hg19) and sf9 genome sequence was produced for analysis of H3K27me3 ChIP-seq dataset obtained using the patient tissue, whereas Drosophila (dm6) genome sequence was used for analysis of H3K27me3 ChIP-seq in SF7761 and SF8628 cells. The mouse (mm9) and Drosophila (dm6) genome sequence was combined and used for analysis of H3K27me3 ChIP-seq in mouse ES cells. Three custom Bowtie2 libraries for combined genome sequence were built by Bowtie2-build. All sequenced reads were aligned against custom library using bowtie2 with default parameters. After alignment, the consistent pair reads were split into each organism. The sequence reads of spike-in chromatin were used to determinate the normalization factor. In order to compare H3K27me3 ChIP-seq read density between *H3F3A* WT or *H3F3A* K27M mutant cells or tissues, the number of reads in each H3K27me3 peak was counted by BEDTools and normalized using the normalization factor.

Different numbers of H3K27me3 peaks were identified among two DIPG lines SF7761 and SF8682 and one primary DIPG tissue. These differences are likely due to different cell lines used, which grow in different media and conditions. However, the difference does not affect our conclusions because we analyzed the H3K27me3 ChIP-seq result at each cell line independently. The analysis from each line supports the same conclusion.

## Accession number

Raw data have been deposited in the GEO database with the series accession GSE94834.

## Acknowledgements

We thank Dr. Songtao Jia and Gary Struhl for critical reading of this manuscript and Dr. Viviane Tabarv for suggestions and support. Dr. Paul Grundy and Paul Goodyer for WT1 plasmids. We thank Michelle Monje for DIPG cell lines and Dr. Suzanne Baker for granting us the access to RNA-seq datasets used in Fig. 4C. These studies were supported by the NIH CA157489 (ZZ). DF was supported by a fellowship from the Fraternal Order of Eagles Cancer Research Fund.

## Additional information

### Funding

| Funder | Grant reference number | Author |
| --- | --- | --- |
| National Institutes of Health | CA204297 | Zhiguo Zhang |

The funders had no role in study design, data collection and interpretation, or the decision to submit the work for publication.

## Author contributions
Dong Fang, Data curation, Formal analysis, Methodology, Writing—original draft, Project administration, Writing—review and editing; Haiyun Gan, Data curation, Software, Formal analysis, Methodology, Writing—review and editing; Liang Cheng, Resources, Validation, Methodology, Writing—review and editing; Jeong-Heon Lee, Resources, Investigation, Methodology, Writing—review and editing; Hui Zhou, Resources, Data curation, Project administration, Writing—review and editing; Jann N Sarkaria, Resources, Investigation, Writing—review and editing; David J Daniels, Resources, Data curation, Supervision, Methodology, Project administration, Writing—review and editing; Zhiguo Zhang, Conceptualization, Resources, Formal analysis, Supervision, Funding acquisition, Writing—original draft, Project administration, Writing—review and editing

## Author ORCIDs
Dong Fang http://orcid.org/0000-0002-9807-5224
Zhiguo Zhang http://orcid.org/0000-0002-9451-2685

## Decision letter and Author response
Decision letter https://doi.org/10.7554/eLife.36696.030
Author response https://doi.org/10.7554/eLife.36696.031

# Additional files

## Supplementary files
• Supplementary file 1. Tables of cell lines, tissues and oligonucleotides used.
DOI: https://doi.org/10.7554/eLife.36696.019
• Transparent reporting form
DOI: https://doi.org/10.7554/eLife.36696.020

## Data availability
Sequencing data have been deposited in GEO under accession codes GSE94834.

The following dataset was generated:

| Author(s) | Year | Dataset title | Dataset URL | Database, license, and accessibility information |
|---|---|---|---|---|
| Zhang Z | 2018 | H3.3K27M mutant proteins reprogram epigenome by sequestering the PRC2 complex to poised enhancers | https://www.ncbi.nlm.nih.gov/geo/query/acc.cgi?acc=GSE94834 | Publicly available at the NCBI Gene Expression Omnibus (accession no: GSE94834) |

The following previously published datasets were used:

| Author(s) | Year | Dataset title | Dataset URL | Database, license, and accessibility information |
|---|---|---|---|---|
| Chan K, Gan H, Zhang Z | 2014 | The histone H3.3K27M mutation in pediatric glioma reprograms H3K27 methylation and gene expression | https://www.ncbi.nlm.nih.gov/geo/query/acc.cgi?acc=GSE61586 | Publicly available at the NCBI Gene Expression Omnibus (accession no: GSE61586) |
| Piunti A, Bartom ET, Shilatifard A | 2016 | Heterotypic nucleosomes and PRC2 drive DIPG oncogenesis | https://www.ncbi.nlm.nih.gov/geo/query/acc.cgi?acc=GSE78801 | Publicly available at the NCBI Gene Expression Omnibus (accession no: GSE78801) |
| Wu G | 2014 | The genomic landscape of diffuse intrinsic pontine glioma and pediatric non-brainstem high-grade glioma | https://www.ebi.ac.uk/ega/studies/EGAS00001000192 | Publicly available at the Electron Microscopy Data Bank (accession no: EGAS0000100192) |

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
