## [Decision Letter]

Thank you for submitting your article "H3.3K27M mutant proteins reprogram epigenome by sequestering the PRC2 complex to poised enhancers" for consideration by *eLife*. Your article has been reviewed by three peer reviewers, including Jerry Workman as the Reviewing Editor and Reviewer #1and the evaluation has been overseen by Jessica Tyler as the Senior Editor. The following individuals involved in review of your submission have agreed to reveal their identity: Hengbin Wang (Reviewer #2); Xiaobing Shi (Reviewer #3).

The reviewers have discussed the reviews with one another and the Reviewing Editor has drafted this decision to help you prepare a revised submission.

Summary:

Histone H3.3 K27 to M mutation occurs frequently in diffuse intrinsic pontine glioma (DIPG), resulting in a significant reduction of H3K27me3 levels; however, H3K27me3 peaks are maintained at certain genomic loci. The mechanism of these changes remains unclear. In this manuscript, the authors report that H3.3 K27M mutant proteins are enriched in highly expressed genes, whereas PRC2 is scarce. An increase of Ezh2 peaks is observed in H3.3 K27M expressing cells and many of these are unique to mutant cells. Despite the presence of PRC2, H3K27me3 is absent from these sites, possibly due to the presence of H3.3K27M mutant proteins at these regions. These regions are enriched for H3K4me1 and identified as poised enhancers. Furthermore, the absence of H3K27M protein at lowly expressing gene promoters (thus have low H3.3K27M protein incorporation) may help retain high levels of H3K27me3. Based on these observations, a model that H3K27M protein sequesters PRC2 to poised enhancers, thus resulting in a reduction of global H3K27me3 and the maintenance of H3K27me3 at lowly expressing gene promoters, is proposed. Finally, they report that tumor suppressor gene WT1 is repressed by H3K27me3 and forced expression of this gene inhibits proliferation of DIPG cells.

Essential revisions:

1) One general concern is that: what is the ratio of poised enhancers and promoters that contain H3K27me3? Can this fully explain the reduction of H3K27me3 in DIPG cells?

2) The genome-wide study in this manuscript clearly shows that H3K27M levels are high at promoters of actively transcribed genes but low at H3K27me3 domains. This is in direct contrast to earlier reports, including work from the same lab, that biochemically Ezh2 is enriched at H3.3K27M mono-nucleosomes than the WT H3.3K27-mononucleosomes isolated from cells. Explanation of such difference is necessary.

3) Ezh2 is redistributed to poised enhancers but not the active promoters where H3K27M proteins are much higher. The authors propose that the PRC2 complex is recruited to poised enhancers independent of H3.3K27M mutant proteins but stays at these sites much longer because of the presence of the H3.3K27M mutant proteins; whereas a lack of initial PRC2 recruiting mechanism to the active gene promoters prevents the PRC2 complex from binding to these sites despite the presence of high levels of H3.3K27M mutant proteins. However, there is not any experimental data to support this hypothesis. It will greatly strengthen the paper if the author can perform some experiments to support this model.

4) In the comparison analyses, some random numbers of genes/peaks are used, e.g., 2000 or 1000 gene promoters for different comparisons. It is not clear how these numbers are chosen. Clear rationale is needed. Or it might be better to divide the genes into high, medium, low groups or something similar for analysis.

5) Figure 1A, 1B, 1D, the signals of H3.3K27M in up-stream of TSS and down-stream of TES are higher than that of gene coding region. Is this a normalization problem? Or does it truly reflect the distribution of H3K27M? Does wild-type H3.3 also behavior like this?

6) Figure 1—figure supplement 1E, it looks like that H3K27me3 is completely abolished in H3.3K27M mutant cells. Can the authors measure the percentage of reduction by titrating down the wild type protein? How about H3K27me2 and H3K27me1? This quantification will also allow the correlation of H3K27me3 reduction with the presence of only half of H3K27me3 peaks in ESCs expressing H3K27M.

7) Figure 2—figure supplement 1B, please explain the figures in detail. How is the conclusion "H3K27me3 levels at all of these peaks in H3.3K27M mutant lines were reduced" drawn?

8) Figure 2C, "the H3.3K27M ChIP-seq read density was significantly higher at Ezh2 peaks unique in H3.3K27M mutant ES cells than those found in both wild type and mutant ES cells". Does this indicate that H3.3K27M incorporate differently into these two regions? Does wild-type H3.3 also have the same behavior?

9) The data in Figure 2F and Figure 2—figure supplement 1C are not exactly the same. Several groups of genes in Figure 2—figure supplement 1C are indeed enriched both for H3.3K27M and H3K27me3, which are not seen in Figure 2F. What is the explanation?

10) The H3K27me3 peaks identified in different cell lines vary a lot, for example, "2,356 H3K27me3 ChIP-seq peaks in SF7761, 5,080 in SF8628 and 12,033 peaks in DIPG patient tissue", "30,295 H3K27me3 ChIP-seq peaks in wild type mouse ES cells, 17,158 peaks in H3.3K27M mutant clone #1 and 16,316 in H3.3K27M mutant clone #2". Is this a technical issue or related to specific cell lines?

11) The manuscript needs to be carefully proof read. There are many grammatical errors throughout.

---

## [Author Response]

Summary:Histone H3.3 K27 to M mutation occurs frequently in diffuse intrinsic pontine glioma(DIPG), resulting in a significant reduction of H3K27me3 levels; however, H3K27me3 peaks are maintained at certain genomic loci. The mechanism of these changes remains unclear. In this manuscript, the authors report that H3.3 K27M mutant proteins are enriched in highly expressed genes, whereas PRC2 is scarce. An increase of Ezh2 peaks is observed in H3.3 K27M expressing cells and many of these are unique to mutant cells. Despite the presence of PRC2, H3K27me3 is absent from these sites, possibly due to the presence of H3.3K27M mutant proteins at these regions. These regions are enriched for H3K4me1 and identified as poised enhancers. Furthermore, the absence of H3K27M protein at lowly expressing gene promoters (thus have low H3.3K27M protein incorporation) may help retain high levels of H3K27me3. Based on these observations, a model that H3K27M protein sequesters PRC2 to poised enhancers, thus resulting in a reduction of global H3K27me3 and the maintenance of H3K27me3 at lowly expressing gene promoters, is proposed. Finally, they report that tumor suppressor gene WT1 is repressed by H3K27me3 and forced expression of this gene inhibits proliferation of DIPG cells.

We thank reviewers for your time reviewing this manuscript and for your insightful comments.

Essential revisions:1) One general concern is that: what is the ratio of poised enhancers and promoters that contain H3K27me3? Can this fully explain the reduction of H3K27me3 in DIPG cells?

We thank the reviewer’s comments. We detected 30295 H3K27me3 peaks in the wild type mouse ES cells. Among them, 5421 peaks are close to promoters and 8846 at the poised enhancers. The number of posited enhancers we identified is similar to the previous report, showing that there are around 11000 poised enhancers in the mouse ES cells (Jaenisch et al., 2010). Therefore, poised enhancers account for about 29% of all H3K27me3 peaks detected. In H3.3K27M mutant cells, we detected far more Ezh2 peaks that are close to these regions. Therefore, we propose that Ezh2 is trapped at poised enhancers for the reduction of H3K27me3 in DIPG cells.

It is possible that other mechanisms also contribute to the global reduction of H3K27me3. Recently, our collaborators used live-cell single-molecule imaging to show that the residence time and search time of Ezh2 was prolonged in H3.3K27M mutant cells but had no effect on its protein level on chromatin (Tatavosian et al., 2018). They also found that Ezh2 is trapped at some loci. However, the identities of these loci were not clear. These results are consistent with our observations. In the revised manuscript, we cited this newly published and discussed these possibilities in the Discussion, subsection “How are the PRC2 complexes redistributed to poised enhancers?”.

2) The genome-wide study in this manuscript clearly shows that H3K27M levels are high at promoters of actively transcribed genes but low at H3K27me3 domains. This is in direct contrast to earlier reports, including work from the same lab, that biochemically Ezh2 is enriched at H3.3K27M mono-nucleosomes than the WT H3.3K27-mononucleosomes isolated from cells. Explanation of such difference is necessary.

We thank the reviewer’s comments. The apparent contradiction of these two results likely results from inadequate explanations. In the genome-wide ChIP-seq study, H3.3K27M proteins are enriched at promoters of actively transcribed genes, like wild type H3.3. This result is consistent with the observation made by Dr. Ali Shilatifard’s laboratory. Moreover, as pointed out in the original manuscript, wild type H3.3 proteins are also enriched at gene promoters and gene bodies of actively transcribed genes. In addition to actively transcribed genes, wild type H3.3 and H3.3K27M are also detected at poised enhancers, while at low levels compared to actively transcribed genes. In our biochemical analysis of Ezh2 that were co-purified with wild type and H3.3K27M mononucleosomes, we expressed H3.3-Flag and H3.3K27M-Flag to a similar level in 293T cells. Assuming that equal amount of H3.3-Flag and H3.3K27M-Flag mutant proteins are found at actively transcribed genes that do not contain Ezh2, the enrichment of Ezh2 at poised enhancers containing H3.3K27M-mononucleosomes as detected by ChIP-seq in this study provided an explanation for the enrichment of Ezh2 on the H3.3K27M-Flag-containing mononucleosomes compared to H3.3-Flag mononucleosomes. We discussed this point in the Results, subsection “Ezh2 is redistributed to poised enhancers without methylating H3K27 at these sites in mutant mouse ES cells” as the following:

“In summary, most H3.3K27M mutant proteins, like wild type H3.3, are enriched at actively transcribed genes where the PRC2 complexes are absent. However, Ezh2, likely other subunits of PRC2 complex, is trapped at poised enhancers where H3.3K27M mutant proteins are detected, albeit at reduced levels compared to actively transcribed genes. This provides an explanation for our previous observation that more Ezh2 proteins were co-purified with H3.3K27M- than wild type H3.3-mononucleosomes (Chan et al., 2013).”

3) Ezh2 is redistributed to poised enhancers but not the active promoters where H3K27M proteins are much higher. The authors propose that the PRC2 complex is recruited to poised enhancers independent of H3.3K27M mutant proteins but stays at these sites much longer because of the presence of the H3.3K27M mutant proteins; whereas a lack of initial PRC2 recruiting mechanism to the active gene promoters prevents the PRC2 complex from binding to these sites despite the presence of high levels of H3.3K27M mutant proteins. However, there is not any experimental data to support this hypothesis. It will greatly strengthen the paper if the author can perform some experiments to support this model.

We agree with the reviewers that it would be a great idea to provide more experimental evidence to support this model. As the reviewers know, how the PRC2 complex is recruited to poised enhancers in mammalian cells is not known. Therefore, it is almost impossible for us to analyze the role of the initial recruitment of the PRC2 complex in the trapping the PRC2 complex to targeted regions. However, this model is supported by our results that PRC2 complex was not detected at actively transcribed genes with high levels of H3.3K27M mutant proteins, and yet PRC2 complex was detected at poised enhancers where it contains low levels of H3K27me3 and PRC2. To address the reviewer’s concern, we validated our findings using ChIP-PCR. Briefly, we performed Ezh2 and H3.3K27M ChIP in wild type and two independent H3.3K27M knocked-in mouse ES cells. The enrichments of Ezh2 and H3.3K27M at three poised enhancers, labeled as enhancer 1 to 3, and one strong PRC2 site, Hoxa1, were analyzed by PCR. We found that Ezh2 was indeed enriched at all poised enhancers in H3.3K27M mutant cells compared to wild type mouse ES cells. Moreover, H3.3K27M mutant proteins were also enriched at these three sites compared to the Hoxa1 site. In addition, Ezh2 was reduced at the Hoxa1 promoter despite low levels of H3.3K27M mutant proteins were detected. These results fully support our original conclusion that the PRC2 complex is redistributed to poised enhancers, which contributes to the reduction of H3K27me3. We included these results in new Figure 2F-G and Results section of the revised manuscript.

Second, as mentioned above, our collaborators recently utilized live-cell single-molecule tracking to determine the impact of H3.3K27M mutant protein on the chromatin binding kinetics of Ezh2. They found that Ezh2 has a prolonged search time in H3.3K27M mutant cells compared to wild type cells, which supports our model that the PRC2 complex stays at poised enhancers much longer, likely due to the presence of the H3.3K27M mutant proteins. In the revised manuscript, we cited this study in the Discussion, subsection “How are the PRC2 complexes redistributed to poised enhancers?”.

Finally, to address this concern further, we point out that “In the future, it will be very interesting to test this model once we know how the PRC2 complex is recruited to poised enhancers.” in the aforementioned Discussion subsection.

4) In the comparison analyses, some random numbers of genes/peaks are used, e.g., 2000 or 1000 gene promoters for different comparisons. It is not clear how these numbers are chosen. Clear rationale is needed. Or it might be better to divide the genes into high, medium, low groups or something similar for analysis.

We thank the reviewers for the suggestions. To address this concern, we divided H3K27me3 peaks into two groups, most reduced and least reduced, based on the average levels of H3K27me3 in both H3.3K27M mutant mouse ES clones, and then performed the same calculation and obtained the same pattern. We replaced original Figure 3 C-I with the new figures and modified the main text and figure legend accordingly.

5) Figure 1A, 1B, 1D, the signals of H3.3K27M in up-stream of TSS and down-stream of TES are higher than that of gene coding region. Is this a normalization problem? Or does it truly reflect the distribution of H3K27M? Does wild-type H3.3 also behavior like this?

We carefully compare the distribution pattern of H3.3K27M with wild type H3.3 in mouse ES cells published in other studies. The higher enrichment of wild type H3.3 at up-stream of TSS and down-stream of TES can be seen in other studies (e.g. Figure 2A of Goldberg et al., 2010 and Figure S1D from Banaszynski et al., 2013). We do not need to perform normalization to perform this analysis as we simply calculate the read density of H3.3K27M at three groups of genes based on expression levels. Therefore, this pattern is unlikely to arise from the normalization problem. Finally, I would like to point out, as described in experimental procedures, we performed H3.3K27M ChIP-seq using MNase-digested chromatin, which could contribute to the high enrichment of H3.3K27M at TSS and TES.

In the revised manuscript, we add the above discussion in the Materials and methods subsection “ChIP-seq data analysis”: “Because we are using the MNase-digested chromatin to perform the ChIP-seq, it may contribute to high enrichment of H3.3K27M at TSS and TES.”

6) Figure 1—figure supplement 1E, it looks like that H3K27me3 is completely abolished in H3.3K27M mutant cells. Can the authors measure the percentage of reduction by titrating down the wild type protein? How about H3K27me2 and H3K27me1? This quantification will also allow the correlation of H3K27me3 reduction with the presence of only half of H3K27me3 peaks in ESCs expressing H3K27M.

We apologize for the misleading representation of Figure 1—figure supplement 1E. According to the reviewer’s suggestions, we analyzed the levels of H3K27me1/me2/me3 in wild type and H3.3K27M mutant cells by titrating cell lysates as requested by the reviewers. It appears that H3K27me1 level in H3.3K27M mutant cells did not change to a detectable degree, whereas H3K27me2 and H3K27me3 were reduced dramatically.

This result is included in Figure 1—figure supplement 1E. The description of this data is in the Results, subsection “H3.3K27M mutant proteins are enriched at highly expressed genes in both DIPG cells as 141 well as mouse ES cells knocked-in with the H3.3K27M mutation”: “To analyze the impact of H3.3K27M mutation on H3K27 methylation, we compared the levels of H3K27me1, H3K27me2 and H3K27me3 in wild type and H3.3K27M mutant cells using different amount of extracts (Figure 1—figure supplement 1E). Compared to wild type cells, H3K27me1 was not affected to a detectable degree in H3.3K27M mutant cells under all these conditions, consistent with published study in DIPG cells (Bender et al., 2013). In contrast, H3K27me2 and H3K27me3 levels, while detectable in high amounts of cell extracts, were reduced dramatically in H3.3K27M mutant cells compared to wild type cells. Moreover, the reduction in H3K27me2/me3 in each of the H3.3K27M mutant ES cell clone is unlikely due to the reduced levels of the PRC2 complex (Figure 1—figure supplement 1F).”

7) Figure 2—figure supplement 1B, please explain the figures in detail. How is the conclusion "H3K27me3 levels at all of these peaks in H3.3K27M mutant lines were reduced" drawn?

We apologize for the unclear presentation of the figure. We add the detailed description in the figure legend. “(B) Compared to WT mouse ES cells, H3K27me3 levels at all peaks in H3.3K27M mutant ES cells were reduced. H3K27me3 ChIP-seq was performed in WT and two H3.3K27M mutant clones spiked-in with chromatin from *Drosophila*. H3K27me3 peaks were identified and cumulative distribution analysis of the Log_2_ ratio of H3K27me3 levels between mutant and WT cells (K27M / WT) was shown (X-axis). The percentage of H3K27me3 peaks at each ratio was shown at the Y-axis. Note that the percentage of H3K27me3 peaks reaches 100 with a Log_2_ ratio of H3K27me3 levels (K27M / WT) less than 0, which indicates the levels of H3K27me3 at all H3K27me3 peaks identified in H3.3K27M mutant ES cells are reduced compared to WT mouse ES cells.”

8) Figure 2C, "the H3.3K27M ChIP-seq read density was significantly higher at Ezh2 peaks unique in H3.3K27M mutant ES cells than those found in both wild type and mutant ES cells". Does this indicate that H3.3K27M incorporate differently into these two regions? Does wild-type H3.3 also have the same behavior?

We thank the reviewers for these insightful questions. To address these questions, we initially planned to perform H3.3 ChIP-seq using antibodies against H3.3 in the mutant cells and compared the incorporation of H3.3K27M and wild type H3.3 in H3.3K27M mutant mouse ES cells. However, the following complications prevent us from taking this approach. First, the incorporation of H3.3K27M mutant proteins would change the gene expression and therefore likely the distribution of wild type H3.3 in H3.3K27M mutant ES cells at some gene loci. Second, the H3.3 antibodies were raised against a peptide to differentiate H3.3 from H3.1. Therefore, the antibodies cannot differentiate wild type H3.3 from H3.3K27M mutant proteins. Importantly, we tested several H3.3 antibodies for ChIP, and it appears that these antibodies also recognize H3.1 to some extent in our hands.

Therefore, to answer the reviewer’s question, we downloaded the published wild type H3.3 ChIP-seq datasets in mouse ES cells based on HA- and EYFP-tagged H3.3 proteins, and then compared the enrichment of wild type H3.3 at these two group of peaks in wild type mouse ES cells. Interestingly, we observed that wild type H3.3 was enriched similarly between Ezh2 peaks unique in H3.3K27M mutant ES cells and those found in both wild type and mutant ES cells. These results indicate that the enrichment of H3.3K27M mutant proteins at Ezh2 peaks unique to H3.3K27M mutant ES cells is different from wild type H3.3. In the revised manuscript, we add these results as Figure 2—figure supplement 1C and described them in the Results, subsection “Ezh2 is redistributed to poised enhancers without methylating H3K27 at these sites in mutant mouse ES cells”. “Interestingly, similar analysis on the published dataset using HA- and EYFP-tagged H3.3 proteins in mouse ES cells showed that wild type H3.3 was enriched to a similar level between these two groups of Ezh2 peaks (Figure 2—figure supplement 1C). The difference in the enrichment of H3.3K27M and wild type H3.3 at these two group of Ezh2 peaks is currently unknown.”

9) The data in Figure 2F and Figure 2—figure supplement 1C are not exactly the same. Several groups of genes in Figure 2—figure supplement 1C are indeed enriched both for H3.3K27M and H3K27me3, which are not seen in Figure 2F. What is the explanation?

We thank the reviewers for the careful examination of these figures. Currently, we do not know the primary reason for the difference. One possible reason is that different antibodies against H3K27me3 were used in these two studies. We used a rabbit monoclonal anti-H3K27me3 antibody (Cell Signaling, Cat.# 9733) and this antibody was validated extensively (Rothbart et al., 2015). Figure 2—figure supplement 1C utilized the H3K27me3 antibodies homemade in Dr. Ali Shilatifard’s laboratory. Moreover, the number of H3K27me3 peaks were also different from these two studies. Using the same cut off value, there were 7610 H3K27me3 peaks in the supplemental figure and 5080 peaks in our study.

In the revised manuscript, we point this out at Figure legend of Figure 2—figure supplement 1C (New Figure 2—figure supplement 1D).

Despite the difference, both studies clearly show that a fraction of Ezh2 co-localized with H3.3K27M mutant proteins.

10) The H3K27me3 peaks identified in different cell lines vary a lot, for example, "2,356 H3K27me3 ChIP-seq peaks in SF7761, 5,080 in SF8628 and 12,033 peaks in DIPG patient tissue", "30,295 H3K27me3 ChIP-seq peaks in wild type mouse ES cells, 17,158 peaks in H3.3K27M mutant clone #1 and 16,316 in H3.3K27M mutant clone #2". Is this a technical issue or related to specific cell lines?

The difference in the number of H3K27me3 peaks likely reflects different cell lines and primary tissue used. While SF7761 and SF8682 are DIPG lines containing H3.3K27M mutation, they grow very differently. For instance, SF7761 cells can only grow like neurosphere in neural stem cell media, whereas SF8628 cells grow adherently in DMEM media. DIPG tissue is different from DIPG cells as it is likely to be heterogeneous. Therefore, it is not surprising that different numbers of H3K27me3 ChIP-seq peaks were identified among these two cell lines and the primary DIPG tissue. Supporting this idea, the number of Ezh2 peaks identified in two H3.3K27M mutant mouse ES clones are very close to each other (about 4.9% difference). Moreover, we also analyzed the H3K27me3 peaks in the ENCODE project GSE49847 and found that there are 18169 H3K27me3 peaks in mouse ES cells and 1730 H3K27me3 peaks in mouse B-cells (CD43-). Therefore, the difference in H3K27me3 peak numbers may arise from different cell lines used.

Importantly, I would like to point out that the difference should not affect our conclusion for two reasons. First, we analyzed H3K27me3 ChIP-seq results in each cell line (Figure 3 F-I) independently and similar conclusions/results were obtained from datasets obtained from these different lines. Second, we utilized the H3K27me3 peaks that commonly found in all three lines (Figure 4—figure supplement 1A) to identify tumor suppressor genes that are silenced in DIPG cells.

In the revised manuscript, we point out in the Materials and methods, subsection “Spike-in chromatin H3K27me3 ChIP-seq analysis”, that “Different numbers of H3K27me3 peaks were identified among two DIPG lines SF7761 and SF8682 and one primary DIPG tissue. These differences are likely due to different cell lines used, which grow in different media and conditions. However, the difference does not affect our conclusions because we analyzed the H3K27me3 ChIP-seq result at each cell line independently. The analysis from each line supports the same conclusion.”

11) The manuscript needs to be carefully proof read. There are many grammatical errors throughout.

We are sorry for our oversight. In the revised manuscript, we edited the manuscript carefully and made every effort to improve the clarity of the manuscript.